

# The partial Bondi gauge: Gauge fixings and asymptotic charges

Marc Geiller[1] and Céline Zwikel[2]

**1** Univ. Lyon, ENS de Lyon, CNRS, Laboratoire de Physique, F-69342 Lyon, France
**2** Perimeter Institute for Theoretical Physics,
31 Caroline Street North, Waterloo, Ontario, Canada N2L 2Y5

## Abstract

In the companion paper [1] we have studied the solution space at null infinity for gravity in the partial Bondi gauge. This partial gauge enables to recover as particular cases and among other choices the Bondi–Sachs and Newman–Unti gauges, and to approach the question of the most general boundary conditions and asymptotic charges in gravity. Here we compute and study the asymptotic charges and their algebra in this partial Bondi gauge, by focusing on the flat case with a varying boundary metric $\delta q_{AB} \neq 0$. In addition to the super-translations, super-rotations, and Weyl transformations, we find two extra asymptotic symmetries associated with non-vanishing charges labelled by free functions in the solution space. These new symmetries arise from a weaker definition of the radial coordinate and switch on traces in the transverse metric. We also exhibit complete gauge fixing conditions in which these extra asymptotic symmetries and charges survive. As a byproduct of this calculation we obtain the charges in Newman–Unti gauge, in which one of these extra asymptotic charges is already non-vanishing. We also apply the formula for the charges in the partial Bondi gauge to the computation of the charges for the Kerr spacetime in Bondi coordinates.



# 1 Introduction

Within the gauge-fixing approach to asymptotic symmetries in general relativity (see [2] for a review), one uses diffeomorphism freedom to write the line element in a preferred gauge, before then imposing physically-motivated boundary conditions. Loosely speaking, the asymptotic symmetries are then the residual diffeomorphisms which preserve both the gauge and the chosen boundary conditions. One may then further distinguish, according to standard terminology, the physical diffeomorphisms having a non-vanishing asymptotic charge from those that are pure gauge and associated with a vanishing charge. This subtle interplay between the choice of gauge-fixing, the boundary conditions, and the resulting asymptotic charges, is at the heart of recent investigations on the asymptotic structure of spacetime, in particular in relation with gravitational waves, holography, and the so-called infrared triangle.

In the companion paper [1] we have approached these questions in a broad setup called the partial Bondi gauge. This gauge follows Bondi's original idea of working in coordinates $(u, r, x^A)$ adapted to the null geodesics of the spacetime [3], but is only a *partial* gauge fixing since it is defined (in four spacetime dimensions) by the three conditions $g^{uu} = 0 = g^{uA}$, and therefore has a residual freedom in the choice of the radial coordinate. Two convenient and often used complete gauge fixings of this partial Bondi gauge are the Bondi–Sachs (BS) gauge [4–8] and the Newman–Unti (NU) gauge [9], in which $r$ is respectively an areal radius and the affine parameter for the outgoing geodesics. Working in the partial Bondi gauge has the advantage of encompassing both the BS and the NU gauge, as well as any other complete gauge fixing corresponding to an arbitrary choice of the radius $r$. The study of such a partial gauge is possible because the three aforementioned conditions on the metric are sufficient to solve the Einstein equations once a radial expansion for the angular metric is chosen. In [1] we have focused on the details of the resolution of the Einstein equations in the partial Bondi gauge, and on the construction of a general solution space containing an arbitrary cosmological constant, logarithmic terms, and a free time-dependent boundary metric. This is all the structure which is allowed by the Einstein equations once a fall-off (in $r^2$) for the transverse metric has been chosen. In the present work we focus instead on the asymptotic symmetries and charges of the partial Bondi gauge.

The charges for asymptotically-flat spacetimes in the Bondi–Sachs (BS) gauge have been studied extensively, starting with the early work of BMS [4–8] and then investigating relaxations of the boundary conditions. This has lead in particular to the introduction of the so-called generalized BMS (GBMS) [10–20] and BMS–Weyl (BMSW) [21, 22] frameworks, in which the BS gauge is supplemented by boundary conditions which allow for an arbitrary

metric on the asymptotic 2-sphere (whose volume is fixed in the case of GBMS and arbitrary for BMSW). While in the structure of the BMS charges the mass and angular momentum can be thought of as respectively paired with super-translations and super-rotations, in BMSW the volume of the induced boundary metric is paired with the Weyl charge. This illustrates how relaxing the gauge and/or the boundary conditions so as to have free functions in the metric can lead to new free symmetry parameters and associated non-vanishing charges. In this work we show that in the partial Bondi gauge[1] at most two new asymptotic charges can exist, and we exhibit complete gauge fixings in which these charges are indeed non-vanishing. At the difference with mass and angular momentum, these new charges have no associated flux-balance laws, and are therefore on a similar footing as the Weyl charge. In fact they appear in the same way as the latter, i.e. via free symmetry parameters in the expansion of the radial asymptotic Killing vector.

Curiously, by contrast with the vast existing literature on asymptotic charges in Bondi–Sachs (BS) gauge, to the best of our knowledge the expression for the asymptotic charges in Newman–Unti (NU) gauge has not appeared previously in metric form. The NU gauge is indeed typically used in the Newman–Penrose formalism [23,24], in which the asymptotic charges have been studied for example in [25–28], but the translation to metric variables is not straightforward and relies on a technical dictionary [29]. Computing the asymptotic charges in the partial Bondi gauge enables to fill this gap and to obtain the charges of the NU gauge in metric variables as a simple limiting case. This has potential applications to the study of the interplay between asymptotic charges and gravitational radiation, since for example when mapping harmonic coordinates to Bondi-type coordinates one is naturally led to the NU gauge and not the BS one [30,31]. Interestingly, a similar situation is also encountered when bringing the Kerr metric in Bondi form [32–34], where one is first naturally led by the diffeomorphism to coordinates referred to by the authors of [32] as "generalized Bondi coordinates", and in terms of which the line element is neither in BS gauge nor in NU gauge. These so-called generalized Bondi coordinates belong to the partial Bondi gauge, and instead of performing an additional diffeomorphism to BS gauge as was done in [12], one can use the general form of the asymptotic charges in the partial Bondi gauge to compute the charges of Kerr at null infinity.

The study of the asymptotic charges in the partial Bondi gauge is also related to important and deep conceptual aspects. Many recent attempts have been made at tackling the question of the most general charges and boundary symmetries in general relativity [35–55], with tentative applications to holography or a quasi-local generalization thereof [56,57]. While for arbitrary boundaries at finite distance it is possible to argue in favor of a preferred "universal corner symmetry" algebra and its charges, little is known about how these quasi-local considerations connect with the study of asymptotic symmetries and their charges. Part of the difficulties in this question lies in the fact that the asymptotic symmetries are better studied with a choice of gauge and boundary conditions, and require to have control over an asymptotic solution space. It is indeed well-known that finding a gauge and boundary conditions which enable to solve the Einstein equations is a very subtle task for which there is unfortunately no general guiding principle and one must often resort to trial-and-error. There are however a few isolated occurrences in which this was achieved successfully and new asymptotic charges were discovered. This is for example the case of BMSW [21], where the study of the asymptotic Weyl charge was precisely motivated by the question of whether finite distance Weyl rescalings have an asymptotic counterpart. Many alternative gauge fixings and bound-

---

[1]More precisely, with respect to the general setup of [1] here we consider a sub-sector in which the cosmological constant is vanishing, there are no logarithmic terms, and the boundary metric is time-independent and only allowed to vary in the transverse directions. In equations, this means that $\Lambda = 0$, $\partial_u q_{AB} = 0$, $U_0^A = 0$, $\beta_0 = 0$, and $\delta q_{AB} \neq 0 \neq \delta\sqrt{q}$.

ary conditions have also been explored successfully in three-dimensional spacetimes, both in Bondi gauge [42, 53, 58–64] and in Fefferman–Graham gauge [36, 37, 65–68]. In light of this discussion, since the partial Bondi gauge enables to solve the Einstein equations and define a very broad solution space [1], it is therefore natural and important to study the associated asymptotic charges. That this is at all possible is in fact surprising and lucky. What the present work reveals is that the partial Bondi gauge generically contains two new charges in addition to those present in BMSW. The associated new symmetry parameters are the two subleading contributions appearing after the Weyl generator in the radial part of the asymptotic vector field, while the corresponding free functions in the solution space are the traces of the two subleading tensors appearing after the leading angular boundary metric. Interestingly, one of the two new asymptotic charges is already present if we simply use the NU gauge as a complete gauge fixing of the partial Bondi gauge. We also exhibit new full gauge fixing conditions, which are differential generalizations of the BS and NU gauge, and in which the two new asymptotic charges survive. The resulting asymptotic symmetry algebra is given by (2.23). We defer the study of its relationship with the corner symmetry algebra to future work.

The outline of this work and of the results is as follows. We begin in section 2 by recalling the results of [1] on the structure of the solution space in partial Bondi gauge. We explain in particular how the partial Bondi gauge contains an infinite amount of free functions of the retarded time $u$, which are the traces of the two-dimensional tensors appearing in the expansion of the angular metric. Completing the partial Bondi gauge to a full gauge fixing leaves only a finite number of these traces as free data on $\mathcal{I}^+$. For example, in BS gauge there is no free trace, while in NU gauge the trace of the first subleading term $C_{AB}$ in the expansion of the angular metric is free. We then study in this same section the asymptotic Killing vectors, and show that in the partial gauge the radial part of the vector field contains an infinite amount of free functions of $(u, x^A)$ in its $1/r$ expansion. Only the first three contributions in this expansion appear in the asymptotic charges. These free functions are paired with data in the solution space: the first free function is the Weyl generator and is paired with the boundary volume $\sqrt{q}$, while the second and third free functions are paired with the traces of $C_{AB}$ and the subleading term $D_{AB}$. We then study the symplectic potential in order to show which sources of flux are turned on in the partial Bondi gauge.

In section 3 we introduce two new gauge fixing conditions, called the differential BS and NU gauges, which enable to completely gauge fix the partial Bondi gauge while keeping an arbitrary *finite* number of the angular traces as free functions. These are gauge fixings in which the corresponding asymptotic symmetry parameters remain free as well, and therefore lead to genuine new non-vanishing and independent charges. We also provide a Carrollian interpretation of these new gauge fixings in terms of the inaffinity and the expansion of the rigging vector.

Section 4 is devoted to the computation of the asymptotic charges, using standard covariant phase space methods. As announced above, we find that when the two-dimensional boundary metric is allowed to fluctuate, i.e. when $\delta q_{AB} \neq 0$, two new asymptotic charges appear, with symmetry generators corresponding to the two subleading terms after the Weyl parameter in the radial part of the vector field. The charges are also divergent (as it is already the case with GBMS), and we show how they can be renormalized. As a consistency check we match our result in a particular case with the GBMS charges found in [20], and also compute the charges for the Kerr solution using the partial Bondi gauge.

In section 5 we study the algebra of the charges. Since this is a very lengthy computation, we perform it in a slightly simplified setup where we choose the boundary metric to be conformal to a fixed two-sphere, with the conformal factor as the only field space variable. We compute the charge algebra using both the Barnich–Troessaert bracket [12] and the recently introduced Koszul bracket [69,70]. With the former we find that the charge algebra contains a

field-dependent two-cocycle, while with the Koszul bracket we find that the field-dependency drops.

We finally give some perspectives for future work in section 6, and defer some lengthy computations required for the charge algebra to appendices.

## 2 The partial Bondi gauge

In this section we gather from [1] the necessary ingredients about the partial Bondi gauge in order to compute the charges and their algebra. We first recall the structure of the solution space and the evolution equations, before studying the asymptotic symmetries, the associated transformation laws, and the symplectic structure.

### 2.1 Solution space

Let us consider coordinates $x^\mu = (u, r, x^A)$, where $u$ is the retarded time, $r$ the radial coordinate and $x^A$ the angular coordinates. The partial Bondi gauge is defined by the three gauge conditions $g^{uu} = 0 = g^{uA}$, or equivalently $g_{rr} = 0 = g_{rA}$. With these gauge conditions the line element takes the form

$$\mathrm{d}s^2 = e^{2\beta}\frac{V}{r}\mathrm{d}u^2 - 2e^{2\beta}\mathrm{d}u\,\mathrm{d}r + \gamma_{AB}(\mathrm{d}x^A - U^A\mathrm{d}u)(\mathrm{d}x^B - U^B\mathrm{d}u), \tag{2.1}$$

where at this stage $V(u, r, x^A)$, $\beta(u, r, x^A)$, and $U^A(u, r, x^B)$ are four unspecified functions of the four spacetime coordinates. In [1] we have solved the Einstein equations using this partial Bondi gauge in a very general setup, containing in particular a non-vanishing cosmological constant, logarithmic terms, and a free time-dependent boundary metric on $\mathcal{I}^+$. Here we will focus instead on a small subsector of this general solution space, as it will be sufficient to reveal the appearance of extra boundary charges. More precisely, we will consider the case of a vanishing cosmological constant, discard logarithmic terms, and freeze all but the transverse part of the boundary metric.

In order to solve the Einstein equations, we first need to choose an expansion and fall-off conditions for the transverse metric. We consider

$$\gamma_{AB} = r^2 q_{AB} + rC_{AB} + D_{AB} + \frac{1}{r}E_{AB} + \mathcal{O}(r^{-2}) = \gamma_{AB} = r^2 q_{AB} + rC_{AB} + \sum_{n=0}^{\infty}\frac{\gamma_{AB}^n}{r^n}, \tag{2.2}$$

and exclude in particular logarithmic terms as mentioned above. We also consider that the leading transverse metric is time-independent, i.e. that $\partial_u q_{AB} = 0$. The vacuum Einstein equations with no cosmological constant can then be solved in a $1/r$ expansion, and we find at leading order that

$$\beta = \frac{\beta_2}{r^2} + \mathcal{O}(r^{-3}) = \frac{1}{32r^2}\big([CC] - 4D\big) + \mathcal{O}(r^{-3}), \tag{2.3a}$$

$$U^A = \frac{U_2^A}{r^2} + \frac{U_3^A}{r^3} + \mathcal{O}(r^{-4}) = \frac{1}{2r^2}\big(\partial^A C - D_B C^{AB}\big) + \frac{N^A}{r^3} + \mathcal{O}(r^{-4}), \tag{2.3b}$$

$$V = -\frac{1}{2}\big(R + \partial_u C\big)r + 2M + \mathcal{O}(r^{-1}). \tag{2.3c}$$

Here $R$ is the Ricci scalar of the metric $q_{AB}$, we have denoted $[CD] := C^{AB}D_{AB}$, and the traces are $C := q^{AB}C_{AB}$ and $D := q^{AB}D_{AB}$. Our boundary conditions such that the induced metric on $\mathcal{I}^+$ depends only on $q_{AB}$ imply that $\beta$ and $U^A$ contain no terms of order $\mathcal{O}(1)$. When comparing

with [1] this means that here we are setting $\beta_0 = 0 = U_0^A$. Finally, the integration constants $M(u, x^A)$ and $N^A(u, x^B)$ are the bare mass and angular momentum aspects.

The form of the solution (2.3) reveals the key feature of the partial Bondi gauge, which is that the various tensors (appart from $q_{AB}$) appearing in the expansion (2.2) have a free trace in $q_{AB}$. This is the reason why the traces $C$ and $D$ appear in the solution at leading order. In the partial Bondi gauge, all the traces are free functions of $(u, x^A)$. As explained at length in [1], it is only when further reducing the partial Bondi gauge to a complete gauge fixing that some of these traces are determined. For example, the Bondi–Sachs (BS) gauge [5,6] is obtained from the partial Bondi gauge by further imposing the determinant condition $\det \gamma_{AB} = r^4 \det q_{AB}^\circ$ where $q_{AB}^\circ$ is a fixed round sphere metric. At leading order, this determinant condition implies that $C = 0$, while at subleading order it gives $2D = [CC]$ and at subsubleading order $E = [CD]$. Alternatively, one may consider the Newman–Unti (NU) gauge [9], which is a complete gauge fixing of the partial Bondi gauge obtained with the additional condition $\beta = 0$. One can see in (2.3a) that this will also determine the traces, starting with $4D = [CC]$, however while still leaving $C$ unconstrained. In this sense, the NU gauge can be considered as weaker than the BS gauge: it fixes all the traces in the expansion of the transverse metric apart from that of $C_{AB}$. Again, we refer the reader to the companion paper [1] for more details about this mechanism. In section 3 we will exhibit even weaker variants of the NU and BS gauges, i.e. complete gauge fixings which leave an arbitrary but finite number of traces undetermined.

Going back to the solution space in partial Bondi gauge, let us now note that in order to avoid $\ln(r)$ terms from appearing in the solution we need to impose the additional condition

$$D_{AB}^{\text{TF}} = \frac{1}{4}CC_{AB}^{\text{TF}} \qquad \Rightarrow \qquad D_{AB} = \frac{1}{2}q_{AB}D + \frac{1}{4}CC_{AB}^{\text{TF}}, \tag{2.4}$$

where $_{\text{TF}}$ denotes the symmetric and trace-free part in $q_{AB}$. This is a generalization to the partial Bondi gauge of the usual condition $D_{AB}^{\text{TF}} = 0$ imposed in Bondi–Sachs gauge in order to remove the logarithmic branches.

Finally, we can turn to the evolution equations for the mass and the angular momentum. These are the $(uu)$ and $(uA)$ Einstein equations. They can be written more compactly in terms of the so-called covariant functionals identified as the leading terms in the Newman–Penrose Weyl scalars [23]. These covariant functionals are given by [1]

$$\mathcal{E}_{AB} := 3E_{AB}^{\text{TF}} + \frac{3}{16}C_{AB}^{\text{TF}}\big([CC] - 4D\big), \tag{2.5a}$$

$$\mathcal{P}_A := -\frac{3}{2}N_A + \frac{3}{32}\partial_A\big(4D - [CC]\big) + \frac{3}{4}C_{AB}\big(D_C C^{BC} - \partial^B C\big), \tag{2.5b}$$

$$\mathcal{M} := M + \frac{1}{16}\partial_u\big(4D - [CC]\big), \tag{2.5c}$$

$$\widetilde{\mathcal{M}} := \frac{1}{8}\big(2D_A D_B - N_{AB}\big)\widetilde{C}_{\text{TF}}^{AB}, \tag{2.5d}$$

$$\mathcal{J}_A := \frac{1}{2}D^B N_{AB} + \frac{1}{4}\partial_A R, \tag{2.5e}$$

$$\mathcal{N}_{AB} := \frac{1}{2}\partial_u N_{AB}. \tag{2.5f}$$

Here the news is $N_{AB} := \partial_u C_{AB}^{\text{TF}}$, and $\widetilde{C}^{AB} := \epsilon^{AC}C_C{}^B$ with $\epsilon^{AB} = \varepsilon^{AB}/\sqrt{q}$ the Levi–Civita tensor and $\varepsilon^{AB}$ the symbol. We can explicitly see through the appearance of the traces $C$ and $D$ how the partial Bondi gauge (and different complete gauge fixings to e.g. BS or NU) affects the various Weyl scalars. In particular, it is interesting to note that in NU gauge, where $4D = [CC]$, the bare mass aspect $M$ appearing in $g_{uu}$ corresponds already to the covariant mass $\mathcal{M}$, and that a similar simplification occurs in $\mathcal{E}_{AB}$ and $\mathcal{P}_A$.

In terms of the covariant functionals (2.5), the evolution equations now take a compact form which can be derived from the Newman–Penrose Bianchi identities. We find[2]

$$\partial_u \mathcal{J}_A = D^B \mathcal{N}_{AB}, \tag{2.6a}$$

$$\partial_u \mathcal{M} = \frac{1}{2} D_A \mathcal{J}^A + \frac{1}{4} C_{AB}^{\mathrm{TF}} \mathcal{N}^{AB}, \tag{2.6b}$$

$$\partial_u \widetilde{\mathcal{M}} = \frac{1}{2} D_A \widetilde{\mathcal{J}}^A + \frac{1}{4} C_{AB}^{\mathrm{TF}} \widetilde{\mathcal{N}}^{AB}, \tag{2.6c}$$

$$\partial_u \mathcal{P}_A = \partial_A \mathcal{M} + \widetilde{\partial}_A \widetilde{\mathcal{M}} + C_{AB}^{\mathrm{TF}} \mathcal{J}^B. \tag{2.6d}$$

It is important to note that, at the end of the day, the traces $C, D, \dots$ which are free in the partial Bondi gauge have no associated equations of motion (or flux-balance laws). These traces therefore represent completely free data.

In summary, the solution space contains data with a completely unspecified $u$ dependency ($C_{AB}$ and all the traces $C, D, \dots$), as well as data constrained by the flux-balance laws ($\mathcal{M}, \mathcal{P}_A, \mathcal{E}_{AB}^{\mathrm{TF}}$, and more generally all the trace-free subleading terms in $\gamma_{AB}$) and the boundary data $q_{AB}$. Now that we have characterized the solution space, we can turn to the study of the asymptotic symmetries.

## 2.2 Asymptotic Killing vectors and transformation laws

The asymptotic Killing vectors are the vector fields which act on the metric while preserving the three partial Bondi gauge conditions as well as the fall-off conditions. Preserving the partial Bondi gauge, i.e. imposing $\mathcal{L}_\xi g_{rr} = 0 = \mathcal{L}_\xi g_{rA}$, implies that the vector field $\xi = \xi^u \partial_u + \xi^r \partial_r + \xi^A \partial_A$ has temporal and angular components given by

$$\xi^u = f, \qquad \xi^A = Y^A + I^A = Y^A - \int_r^\infty \mathrm{d}r' \, e^{2\beta} \gamma^{AB} \partial_B f = Y^A - \frac{\partial^A f}{r} + \frac{C^{AB} \partial_B f}{2r^2} + \mathcal{O}(r^{-3}), \tag{2.7}$$

with $\partial_r f = 0 = \partial_r Y^A$. From this, we can already deduce the transformation laws

$$\delta_\xi \gamma_{AB} = \left( f \partial_u + \mathcal{L}_Y + \mathcal{L}_I \right) \gamma_{AB} + \xi^r \partial_r \gamma_{AB} - \gamma_{(AC} U^C \partial_{B)} f, \tag{2.8a}$$

$$\delta_\xi \ln \gamma = \left( f \partial_u + \xi^r \partial_r \right) \ln \gamma + 2 \mathcal{D}_A \xi^A - 2 U^A \partial_A f, \tag{2.8b}$$

$$\delta_\xi g_{ur} = -e^{2\beta} \left( 2 \xi^\mu \partial_\mu \beta + \partial_r \xi^r + U^A \partial_A f + \partial_u f \right), \tag{2.8c}$$

where $\mathcal{D}_A$ is the covariant derivative with respect to the metric $\gamma_{AB}$, and where the determinant of this angular metric is $\gamma := \det \gamma_{AB}$. Note that we also have $\sqrt{-g} = e^{2\beta} \sqrt{\gamma}$.

At this stage the radial part of the vector field is still completely arbitrary. However, if we now require that the asymptotic Killing vector preserves the expansion (2.2) of the angular metric, we find that this constrains the radial component to be of the form

$$\xi^r = rh + \sum_{n=0}^\infty \frac{\xi_n^r}{r^n}. \tag{2.9}$$

Importantly, the functions $h(u, x^A)$ and $\xi_n^r(u, x^A)$ are at the moment all free since we are in the partial Bondi gauge. The reduction of this partial gauge to a complete gauge (e.g. to BS or NU gauge, or to a more general choice) will later on determine how many functions remain free in this expansion for $\xi^r$. With the form (2.9) of the radial component, we can now compute

$$\delta_\xi \ln \sqrt{q} = D_A Y^A + 2h, \tag{2.10}$$

---

[2]We omit the evolution equation for $\mathcal{E}_{AB}$ since it will not play any role here.

which shows that $h$ parametrizes the Weyl transformations. Since the boundary sources have been fixed by the boundary conditions $\beta_0 = 0$ and $U_0^A = 0$, we also have $h = -\partial_u f$ and $\partial_u Y^A = 0$, and for a time-independent boundary metric $\partial_u q_{AB} = 0$ the above transformation law tells us that $\partial_u h = 0$ as well. Indeed, had we kept $\beta_0 \neq 0$ and $U_0^A \neq 0$, we would have found the transformation laws

$$\delta_\xi \beta_0 = \left(f\,\partial_u + \mathcal{L}_Y\right)\beta_0 + \frac{1}{2}\left(h + \partial_u f + U_0^A \partial_A f\right), \tag{2.11a}$$

$$\delta_\xi U_0^A = \left(f\,\partial_u + \mathcal{L}_Y + \partial_u f + U_0^C \partial_C f\right)U_0^A - \partial_u Y^A, \tag{2.11b}$$

which does indeed show that setting $\beta_0 = 0 = U_0^A$ leads to the constraints $h = -\partial_u f$ and $\partial_u Y^A = 0$.

In summary, in the partial Bondi gauge the asymptotic Killing vector has components given by (2.7) and (2.9), together with the conditions $h = -\partial_u f$ and $\partial_u Y^A = 0 = \partial_u h$.

At this point, it is convenient to perform a field-dependent redefinition of the free functions appearing in (2.9). For this, we write

$$\xi_0^r = k + \frac{1}{2}\Delta f, \qquad \xi_1^r = \ell + U_2^A \partial_A f = \ell - \frac{1}{2}D_A C^{AB} \partial_B f + \frac{1}{2}\partial^A C \partial_A f. \tag{2.12}$$

We are therefore trading the free functions $\xi_0^r$ and $\xi_1^r$ for $k$ and $\ell$. These redefinitions are such that $k(u, x^A)$ and $\ell(u, x^A)$ are free functions whose limits to BS and NU gauge correspond simply to

$$k\big|_{\text{BS}} = 0, \qquad k\big|_{\text{NU}} = (\text{free}), \qquad \ell\big|_{\text{BS}} = -\frac{1}{4}C^{AB}D_A \partial_B f, \qquad \ell\big|_{\text{NU}} = 0. \tag{2.13}$$

Accordingly, one should remember that when reducing the partial Bondi gauge to the BS or NU gauge we have

$$C\big|_{\text{BS}} = 0, \qquad C\big|_{\text{NU}} = (\text{free}), \qquad D\big|_{\text{BS}} = \frac{1}{2}[CC], \qquad D\big|_{\text{NU}} = \frac{1}{4}[CC]. \tag{2.14}$$

Let us also note that in the BS and NU gauges we have

$$\xi_{n\geq 1}^r\big|_{\text{BS}} = \frac{1}{2}\left(U_{n+1}^A \partial_A f - D_A I_{n+1}^A\right), \qquad \xi_{n\geq 1}^r\big|_{\text{NU}} = \frac{1}{n}U_{n+1}^A \partial_A f, \tag{2.15}$$

where $I_n^A$ is the term of order $r^{-n}$ in the expansion of $I^A$ defined in (2.7), and similarly for $U_n^A$ appearing in the solution (2.3b).

With all these ingredients we can now compute the variations of the components of the transverse metric (2.2). Focusing separately on the trace-free parts and the traces, we first find

$$\delta_\xi q_{AB} = \left(\mathcal{L}_Y + 2h\right)q_{AB}, \tag{2.16a}$$

$$\delta_\xi C_{AB}^{\text{TF}} = \left(f\,\partial_u + \mathcal{L}_Y + h\right)C_{AB}^{\text{TF}} - 2(D_A \partial_B f)^{\text{TF}}, \tag{2.16b}$$

$$\delta_\xi D_{AB}^{\text{TF}} = \left(f\,\partial_u + \mathcal{L}_Y\right)D_{AB}^{\text{TF}} + \frac{1}{2}\left(2k + \Delta f\right)C_{AB}^{\text{TF}} - (C_{AC}D^C \partial_B f)^{\text{TF}}, \tag{2.16c}$$

where we have used $\partial_u q_{AB} = 0$. The variations of the traces are

$$\delta_\xi C = \left(f\,\partial_u + \mathcal{L}_Y - h\right)C + 4k, \tag{2.17a}$$

$$\delta_\xi D = \left(f\,\partial_u + \mathcal{L}_Y - 2h\right)D + 4\ell + kC - C_{\text{TF}}^{AB}D_B \partial_A f, \tag{2.17b}$$

and that of the news $N_{AB} = \partial_u C_{AB}^{\text{TF}}$ is

$$\delta_\xi N_{AB} = \left(f\,\partial_u + \mathcal{L}_Y\right)N_{AB} + 2\left(D_A \partial_B h\right)^{\text{TF}}. \tag{2.18}$$

For the transformation of the bare mass and the covariant mass (2.5c) we find

$$\delta_\xi M = \left(f\,\partial_u + \mathcal{L}_Y - 3h\right)M + \mathcal{J}^A\partial_A f - \partial_u \ell, \tag{2.19a}$$

$$\delta_\xi \mathcal{M} = \left(f\,\partial_u + \mathcal{L}_Y - 3h\right)\mathcal{M} + \mathcal{J}^A\partial_A f, \tag{2.19b}$$

while for the covariant momentum (2.5b) we find

$$\delta_\xi \mathcal{P}_A = \left(f\,\partial_u + \mathcal{L}_Y - 2h\right)\mathcal{P}_A + 3\left(\mathcal{M}\partial_A + \widetilde{\mathcal{M}}\tilde\partial_A\right)f. \tag{2.20}$$

The transformation laws (2.17) illustrate the one-to-one correspondence between free traces in the partial Bondi gauge and free functions in the asymptotic Killing vectors: if $C$ is free then $k$ is free, and if $D$ is free then $\ell$ is free. This is the continuation of what happens already at leading order in the transverse metric with $\sqrt{q}$, since when the latter is allowed to vary then $h$ is a free function in the asymptotic Killing vector and the Weyl transformations are unleashed. Moreover, since $C$ and $D$ have an arbitrary time-dependency, this is also the case for $k$ and $\ell$. The transformation law (2.17a) has been used to argue that one can use the transformation generated by $k$ to set $C = 0$ [25–27]. In NU gauge this amounts to fixing the origin of the affine parameter. However, we will see shortly that the symmetry parameter $k$ appears in the asymptotic charges. This indicates that one should see the symmetry generated by $k$ as physical and not pure gauge. In fact, we will see that also $\ell$ appears in the asymptotic charges and should therefore be regarded as physical, while the rest of the expansion (2.9) for $n \geq 2$ is pure gauge.

Finally, we close this analysis of the symmetries with the computation of the algebra of vector fields. With the field redefinitions performed above and the knowledge of the various transformation laws, we can compute the adjusted bracket to find

$$\left[\xi(f_1,k_1,\ell_1,Y_1),\xi(f_2,k_2,\ell_2,Y_2)\right]_* = \left[\xi(f_1,k_1,\ell_1,Y_1),\xi(f_2,k_2,\ell_2,Y_2)\right] - \left(\delta_{\xi_1}\xi_2 - \delta_{\xi_2}\xi_1\right)$$
$$= \xi(f_{12},k_{12},\ell_{12},Y_{12}), \tag{2.21}$$

where

$$f_{12} = f_1\partial_u f_2 + Y_1^A\partial_A f_2 - \delta_{\xi_1}f_2 - (1 \leftrightarrow 2), \tag{2.22a}$$

$$k_{12} = f_1\partial_u k_2 + Y_1^A\partial_A k_2 + (\partial_u f_1)k_2 - \delta_{\xi_1}k_2 - (1 \leftrightarrow 2), \tag{2.22b}$$

$$\ell_{12} = f_1\partial_u \ell_2 + Y_1^A\partial_A \ell_2 + 2(\partial_u f_1)\ell_2 - \delta_{\xi_1}\ell_2 - (1 \leftrightarrow 2), \tag{2.22c}$$

$$Y_{12}^A = Y_1^B\partial_B Y_2^A - \delta_{\xi_1}Y_2^A - (1 \leftrightarrow 2). \tag{2.22d}$$

The remarkable result is that when $(f,k,\ell,Y^A)$ are field-independent we therefore obtain an algebra (as opposed to an algebroid). This is due to the non-trivial field-dependent redefinition performed in (2.12). Decomposing the $u$ part of the asymptotic Killing vector field as $f(u,x^A) = T(x^A) - uh(x^A)$, where $T$ is a supertranslation, we find that the commutation relations (2.22) encode the algebraic structure of

$$\left(\left(\mathrm{Diff}(S^2) \oplus \mathbb{R}_T\right) \oplus \mathbb{R}_h\right) \oplus \mathbb{R}^2_{k,\ell}. \tag{2.23}$$

The main result of this work will be to show in section 4 that when $\delta q_{AB} \neq 0$ and $\delta q \neq 0$ this whole algebra is associated with non-vanishing asymptotic charges.

## 2.3 Symplectic potential

The study of the symplectic potential is useful for two main reasons. First, its finite part gives information about the conjugate pairs on $\mathcal{I}^+$ and about the sources of flux and non-integrability

in the charges. Second, its divergent part can be used to renormalize the divergencies in the charges using a corner term. The (pre-)symplectic potential of interest for us is that arising from the Einstein–Hilbert Lagrangian, namely

$$\theta^\mu = \sqrt{-g}\left(g^{\alpha\beta}\Gamma^\mu_{\alpha\beta} - g^{\mu\alpha}\Gamma^\beta_{\alpha\beta}\right). \tag{2.24}$$

The time component of this potential has an expansion of the form $\theta^u = r\theta^u_{\text{div}} + \theta^u_0 + \mathcal{O}(r^{-1})$, with

$$\theta^u = 2r\delta\sqrt{q} - \frac{1}{2}\left(\sqrt{q}\,q^{AB}\delta C^{\text{TF}}_{AB} - \delta\sqrt{q}\,C\right) + \mathcal{O}(r^{-1}). \tag{2.25}$$

The radial part has a similar expansion $\theta^r = r\theta^r_{\text{div}} + \theta^r_0 + \mathcal{O}(r^{-1})$, where the divergent piece is

$$\theta^r_{\text{div}} = \frac{1}{2}\partial_u\left(\sqrt{q}\,q^{AB}\delta C^{\text{TF}}_{AB} - \delta\sqrt{q}\,C\right) - \delta\left(\sqrt{q}R\right). \tag{2.26}$$

One can note that this divergent contribution is the sum of a total variation and a so-called corner term, and in particular that we have

$$\theta^r_{\text{div}} = -\partial_u\theta^u_0 - \delta(\sqrt{q}R). \tag{2.27}$$

This can be traced back to the fact that *on-shell* we have $\delta L = \partial_u\theta^u + \partial_r\theta^r + \partial_A\theta^A$, which shows that the divergent part of $\theta^r$ is related to the leading corner term in $\theta^u$. The latter will be used in section 4.2 to renormalize the charges.

Finally, in terms of the news $N_{AB} = \partial_u C^{\text{TF}}_{AB}$ the finite part of the radial component of the potential is given by

$$\begin{aligned}
\theta^r_0 = {} & \frac{1}{4}\sqrt{q}\,q^{AB}\delta\left(RC^{\text{TF}}_{AB} - 2D_A D^C C^{\text{TF}}_{BC} + D_A\partial_B C\right) + \frac{1}{2}\sqrt{q}\,N_{AB}\delta C^{AB}_{\text{TF}} + \frac{1}{4}\sqrt{q}\,\delta CR \\
& + \delta\sqrt{q}\left(2M - \frac{1}{2}D_A\left(\partial^A C - D_B C^{AB}\right) + \frac{1}{2}C^{AB}\partial_u C_{AB} - \frac{1}{2}C\partial_u C\right) \\
& + \delta\left(2\sqrt{q}\,M - \frac{1}{2}\sqrt{q}\,C^{AB}\partial_u C_{AB} - \frac{1}{4}\sqrt{q}\,C\left(R - \partial_u C\right)\right) \\
& + \frac{1}{4}\sqrt{q}\,\partial_u\left(\frac{1}{4}\delta\left(4D - 3[CC]\right) + 2q^{AB}\delta D_{AB}\right) \\
& + \frac{1}{2}\sqrt{q}\,D_A\left(\left(\partial^A C - D_B C^{AB}\right)\delta\ln\sqrt{q}\right). \tag{2.28}
\end{aligned}$$

In addition to the usual Ashtekar–Streubel contribution [71], this symplectic potential contains the contributions arising from the fact that $\delta q_{AB} \neq 0$ and $\delta q \neq 0$, and also from the use of the partial Bondi gauge (through the appearance of $C$ and $D$).

## 3 Complete gauge fixings

We have seen so far that the partial Bondi gauge contains a varying volume element $\sqrt{q}$ and unspecified traces $C, D, \ldots$, and that, correspondingly, the radial part of the asymptotic Killing vector contains and infinite amount of free functions $h, k, \ell, \ldots$ appearing in the expansion (2.9). In section 4 we will show that, generically, when working in the partial Bondi gauge the three functions $h, k, \ell$ actually appear in the asymptotic charges. A natural question is therefore whether this statement remains true, in some form, when completing the partial Bondi gauge to obtain a full gauge fixing with four gauge conditions.

A different way to look at this question is to ask whether there exist complete gauge fixings in which some of the traces in the expansion (2.2) remain free, such that the parameters $h, k, \ell$ themselves remain free. In the BS gauge, we already know that when keeping the volume element $\sqrt{q}$ free the Weyl parameter $h$ is free (by virtue of (2.10)) and appears in the asymptotic charges [21]. The NU gauge, by extension, allows to have both $\sqrt{q}$ and $C$ free at the level of the solutions, and therefore both $h$ and $k$ free at the level of the symmetries. We will now exhibit generalizations of the NU and BS gauge conditions for which an arbitrary finite number of traces can be free. In particular, this contains the cases where both $C$ and $D$ are free, which are full gauge fixings in which both parameters $k$ and $\ell$ appear in the asymptotic charges. For reasons which will become clear in a moment, we call these generalized gauge conditions the *differential* Newman–Unti and Bondi–Sachs gauges. We will conclude this section with a Carrollian interpretation of the new gauges.

## 3.1 Differential Newman–Unti gauges

In order to complete the gauge fixing of the partial Bondi gauge (2.1), let us consider for a given integer $b \in \mathbb{N}$ the differential gauge condition

$$\partial_r^b(r^{b-1}\beta) = \partial_r^b\left(r^{b-1}\ln\sqrt{-g_{ur}}\right) \overset{!}{=} 0. \tag{3.1}$$

This is solved for $b > 0$ by an expansion of the form

$$\beta = \sum_{n=0}^{b-1} \frac{\beta_n}{r^n}, \tag{3.2}$$

which shows that for a fixed $b$ we obtain an extension of the NU gauge where a finite number of terms in the radial expansion of $\beta$ are non-vanishing. More precisely, the family of gauge conditions (3.1) encompasses the following cases:

∗ For $b = 0$ we obtain the usual NU gauge where the gauge condition is setting $\beta = 0$ at every order in $1/r$. All the traces in (2.2) appart from that of $C_{AB}$ are then fixed by the vanishing $1/r$ expansion of the solution (2.3a) to the $(rr)$ Einstein equation.

∗ For $b = 1$ the gauge condition (3.1) becomes $\partial_r\beta = 0$, which implies that $\beta = \beta_0(u, x^A)$. This is the relaxation of the NU gauge already considered in [1] in order to describe a free boundary metric on $\mathcal{I}^+$. Recall however that here we have chosen boundary conditions such that $\beta_0 = 0$ regardless of the gauge condition. In addition, one can note that the gauge condition $\partial_r\beta = 0$ obtained with $b = 1$ preserves the geometrical meaning of the NU gauge, which is to take the radial coordinate $r$ as the affine parameter for $\partial_r$. This follows from the fact that the inaffinity of $\partial_r$ is $2\partial_r\beta$ [1].

∗ For $b = 2$ we obtain a gauge in which $\beta = \beta_0(u, x^A) + \beta_1(u, x^A)/r$. However, it turns out that the vacuum $(rr)$ Einstein equation actually enforces the condition $\beta_1 = 0$. This implies that the gauges obtained with $b = 1$ and $b = 2$ are equivalent on-shell.

∗ For $b \geq 3$ we obtain new gauges which extend the NU gauge $b = 0$ by allowing a finite number of consecutive terms in the expansion of $\beta$ to be non-vanishing. In terms of free functions in the solution space, one should recall that already $b = 0$ allows to have a free trace $C$. Taking $b = 1$ or $b = 2$ then enables to have $\beta_0$ itself as a free function (and we have discarded this option with our choice of boundary conditions). Taking $b = 3$ then leads to a free trace $D$, and more generally $b \geq 3$ leads to free traces up to $q^{AB}\gamma_{AB}^{b-3}$ in the expansion (2.2).

Now that we understand how the gauge conditions (3.1) affect the solution space by allowing for a finite number of traces to survive as free functions, we want to study what happens at the level of the symmetries. We expect that allowing for more undetermined functions in

the solution space will lead to more freedom in the asymptotic Killing vector. Indeed, one can see from (2.11a) that keeping $\beta_0$ free leads to an arbitrary time dependency in $f$, while instead setting $\beta_0 = 0$ (as we have done with our boundary conditions) leads to $\partial_u f = -h$. With the transformation laws (2.17), we then see that keeping the traces $C$ and $D$ unspecified leads to the arbitrary symmetry parameters $k$ and $\ell$ in the radial part of the asymptotic Killing vector field. Let us now explain the general mechanism behind this observation for $b > 0$. For this, note that preserving the gauge condition (3.1) requires for the vector field to satisfy the equation

$$\frac{1}{2}\partial_r^b\big(r^{b-1}(g_{ur})^{-1}\delta_\xi g_{ur}\big) = 0,\tag{3.3}$$

where $\delta_\xi g_{ur}$ is given by (2.8c). Explicitly, this amounts to imposing

$$\frac{1}{2}\partial_r^b\big(r^{b-1}\big(2f\,\partial_u\beta + 2\xi^A\partial_A\beta + 2\xi^r\partial_r\beta + \partial_r\xi^r + U^A\partial_A f + \partial_u f\big)\big) = 0\,.\tag{3.4}$$

This equation is solved by a radial component $\xi^r$ which satisfies

$$2\xi^r\partial_r\beta + \partial_r\xi^r = \bar{\xi}^r - \big(2f\,\partial_u\beta + 2\xi^A\partial_A\beta + U^A\partial_A f + \partial_u f\big),\qquad \bar{\xi}^r = \sum_{n=0}^{b-1}\frac{\bar{\xi}^r_{n-1}}{r^n},\tag{3.5}$$

where the expansion of $\bar{\xi}^r$ is in terms of $b$ radial integration constants $\bar{\xi}^r_n(u, x^A)$. At the end of the day, the solution for $\xi^r$ can be put in the form[3]

$$\xi^r = k + \frac{1}{2}\Delta f + e^{-2\beta}\left(\int_r^\infty \mathrm{d}r'\, e^{2\beta}\big(-\bar{\xi}^r + 2f\,\partial_u\beta + 2\xi^A\partial_A\beta + U^A\partial_A f + \partial_u f\big)\right),\tag{3.6}$$

where we have used the freedom of redefining the radial integration constant $k(u, x^A)$ in a field-dependent manner so as to obtain $\xi^r_0$ as in (2.12). We should recall that we have $\beta_0 = 0$ because of our choice of boundary condition, and $\beta_1 = 0$ because of the vacuum Einstein equations.

The above calculation shows that the final expression (3.6) for the radial vector field preserving the gauge condition (3.1) contains $(b+1)$ free functions of $(u, x^A)$, which are therefore candidates for new symmetry parameters. Expanding the above solution for the radial vector field for various values of $b$, we find

$$(\xi^r)_{b=1} = rh + k + \frac{1}{2}\Delta f + \mathcal{O}(r^{-1}),\tag{3.7a}$$

$$(\xi^r)_{b=2} = rh + k + \frac{1}{2}\Delta f + (\ln r)\bar{\xi}^r_0 + \mathcal{O}(r^{-1}),\tag{3.7b}$$

$$(\xi^r)_{b=3} = rh + k + \frac{1}{2}\Delta f + (\ln r)\bar{\xi}^r_0 + \frac{1}{r}\big(\ell + U_2^A\partial_A f\big) + \mathcal{O}(r^{-2}),\tag{3.7c}$$

where the free functions $h(u, x^A)$ and $\ell(u, x^A)$ have been obtained respectively from $\bar{\xi}^r_{-1}(u, x^A)$ and $\bar{\xi}^r_1(u, x^A)$ by field-dependent redefinitions so as to be consistent with (2.12). Importantly, in this expansion of $\xi^r$ for various values of $b$ (i.e. for the different gauge choices), the first $(b+1)$ terms contain independent and undetermined integration constants, and the subleading terms which have been omitted contain no new free functions. The freedom in defining a gauge with a choice of $b$ therefore has the consequence of making completely arbitrary the terms up to order $\mathcal{O}(r^{2-b})$ in $\xi^r$, which are the candidates for new symmetry parameters. One should

---

[3]Note that the bound in the integral should not be taken to be $\infty$ for the first terms in $r$ in order to avoid $r$ and $\ln(r)$ divergencies.

note that for $b \geq 2$ if $\bar{\xi}_0^r \neq 0$ the action of $\xi$ on the metric creates logarithmic terms at order $\mathcal{O}(r)$, i.e. at the same order as the shear. From now on we will therefore set $\bar{\xi}_0^r = 0$ in order to prevent the generation of such terms and preserve our choice of fall-offs for the transverse metric (such log terms at the order of the shear were considered in [72–74]).

Let us emphasize once again that only $\ell$ requires the use of the generalized gauge (3.1) with $b \geq 3$ in order to be a free function. By contrast, $h$ and $k$ are naturally present as free symmetry parameters already with the standard NU gauge condition $\beta = 0$. One could then argue that $\ell$ is appearing because the gauge conditions (3.1) with $b \geq 3$ are differential, and therefore "too weak". However, this is not different from what happens when relaxing the determinant condition $\gamma = r^4 q^\circ$ and considering instead the differential condition $\partial_r(\gamma/r^4) = 0$ in order to access the Weyl transformations $h$ in BS gauge [13, 21]. In fact, instead of the NU-type differential gauge condition (3.1), one could consider a generalized differential BS gauge condition with similar properties at the level of the solution space and the radial vector field.

## 3.2 Differential Bondi–Sachs gauges

In the BS gauge it is the determinant condition which is responsible for the determination of the traces of the tensors appearing in (2.2). If we want that some of these traces remain free in the solution space, we therefore need to relax the determinant condition. For this, one can use formula (D.10b) of [1], which gives the expansion of the determinant $\gamma$ in terms of the traces of the tensors appearing in (2.2). One can then write a BS analogue of (3.1) in the form

$$\partial_r^b \left( r^{b-3} \sqrt{\gamma} \right) = \partial_r^b \left( \sqrt{q}\, r^{b-1} \left[ 1 + \frac{C}{2r} + \frac{1}{4r^2} \left( 2D - [CC^{\mathrm{TF}}] \right) + \mathcal{O}(r^{-3}) \right] \right) \overset{!}{=} \frac{1}{r} \partial_r^b \sqrt{q^\circ}, \quad (3.8)$$

where the term $\mathcal{O}(r^{-3})$ contains the trace $E = q^{AB} E_{AB}$ and so on, and where $q^\circ$ is the determinant of a fixed boundary metric. This family of gauge conditions indexed by $b$ encompasses the following cases:

∗ For $b = 0$ we obtain the standard BS determinant condition $\sqrt{\gamma} = r^2 \sqrt{q^\circ}$, which implies at leading order $C = 0$, at subleading order $2D = [CC]$, and so on. One can then see from (2.8a) and (2.8b) that the radial vector field preserving preserving both (2.2) and the gauge condition contains no free functions.

∗ For $b = 1$ we obtain the differential BS condition $\partial_r(\sqrt{\gamma}/r^2) = 0$. This condition still starts at leading order with $C = 0$, and therefore does not allow to obtain a free trace in (2.2). However, with this gauge condition the boundary metric $q_{AB}$ is allowed to differ from the fixed metric $q_{AB}^\circ$. We then obtain from (2.8b) that $\xi^r$ can contain a radial integration constant, which is the free function $h(u, x^A)$ allowing for the Weyl rescalings of $q_{AB}$ [1, 13, 21]. As explained below (2.10), if $q_{AB}$ is chosen as time independent we then have that $\partial_u h = 0$ as well. Once again, this illustrates the link between the introduction of a new freedom in the solution space and the appearance of a new symmetry parameter.

∗ For $b = 2$ we naturally obtain that the gauge condition does not constrain $C(u, x^A)$, which therefore survives the gauge fixing and remains a free function. As expected, the radial vector field preserving this differential gauge condition now contains a free function $k(u, x^A)$. The construction can then be extended to $b \geq 3$ in a similar fashion. One can notice that there is a (innocent) "mismatch" of the order $b$ in the properties of the differential NU and BS gauge conditions. For example, $b = 0$ allows to have a free trace $C$ in the NU gauge, but for the differential BS gauge this requires to take $b = 2$.

Now that we have explained how a finite number of free functions in the expansion (2.9) can survive the full gauge fixing of the partial Bondi gauge with the fourth gauge conditions (3.1) or (3.8), we are going to show in section 4 that at most the first three functions $h, k, \ell$ can appear in the asymptotic charges. At the end of the day, according to the standard classification

the only relevant criterion at this stage is whether the symmetry parameters are associated with vanishing asymptotic charges or not.

### 3.3  Carrollian interpretation

Before computing the charges, let us say a word about the geometrical and Carrollian interpretation of the new differential NU and BS gauges (3.1) and (3.8). Future null infinity $\mathcal{I}^+$, being a null hypersurface, comes equipped with a weak Carrollian structure [49, 62, 75–77]. This consists of a degenerate metric $q^{\mathcal{I}}$ and a nowhere-vanishing null vector $\ell^{\mathcal{I}}$ such that $q^{\mathcal{I}} \cdot \ell^{\mathcal{I}} = 0$. Explicitly, these are given by

$$q^{\mathcal{I}} = q^{\mathcal{I}}_{ab} \mathrm{d}x^a \mathrm{d}x^b = q_{AB}\big(\mathrm{d}x^A - U_0^A \mathrm{d}u\big)\big(\mathrm{d}x^B - U_0^B \mathrm{d}u\big), \qquad \ell^{\mathcal{I}} = e^{-2\beta_0}\big(\partial_u + U_0^A \partial_A\big), \quad (3.9)$$

where the coordinates on $\mathcal{I}^+$ are $x^a = (u, x^A)$, and where we are allowing here for generality to have $\beta_0 \neq 0$ and $U_0^A \neq 0$. This Carrollian structure represents intrinsic boundary data to $\mathcal{I}^+$. In order to probe the bulk of spacetime we need to introduce a transverse quantity, the so-called Ehresmann connection form $\underline{k}^{\mathcal{I}}$, which is dual to $\ell^{\mathcal{I}}$ and given here by

$$\underline{k}^{\mathcal{I}} = -e^{2\beta_0} \mathrm{d}u, \qquad \ell^{\mathcal{I}} \cdot \underline{k}^{\mathcal{I}} = 1. \quad (3.10)$$

This form $\underline{k}^{\mathcal{I}}$ can then be extended into the bulk as a null rigging vector such that its projection to $\mathcal{I}^+$ is $\underline{k}^{\mathcal{I}}$. We take this extension to be simply given by

$$\underline{k} = \underline{k}_\mu \mathrm{d}x^\mu = -e^{2\beta} \mathrm{d}u. \quad (3.11)$$

The inaffinity parameter of the corresponding vector $k^\mu = g^{\mu\nu}\underline{k}_\nu = \delta_r^\mu$ is then defined via $\nabla_k k = \kappa k$ and turns out to be given by $\kappa = 2\partial_r \beta$. In terms of this inaffinity the differential NU gauge condition (3.1) is therefore

$$\partial_r^{b-1}(r^b \kappa) \overset{!}{=} 0. \quad (3.12)$$

Similarly, the differential BS gauge condition (3.8) can also be characterized by a geometrical quantity associated to the rigging vector $k$, namely its expansion scalar

$$\Theta_k = \partial_r \ln \sqrt{\gamma} = \frac{3-b}{r} + \partial_r \log\Big(r^{b-3}\sqrt{\gamma}\Big). \quad (3.13)$$

The strict BS determinant condition with $b = 0$, i.e. $\sqrt{\gamma} = r^2 \sqrt{q^\circ}$, gives $\Theta_k = 2/r$. This explains why in this case the radial coordinate is an areal radius. For different values of $b \geq 1$ we can then rewrite (3.8) in terms of the expansion to find e.g.

$$b = 1: \qquad \Theta_k = \frac{2}{r}, \quad (3.14a)$$

$$b = 2: \qquad \partial_r \Theta_k - \frac{2}{r}\Theta_k + \Theta_k^2 + \frac{2}{r^2} = 0 \qquad \Rightarrow \qquad \Theta_k = \frac{1}{r} + \frac{1}{r + \tau_1}, \quad (3.14b)$$

$$b = 3: \qquad \partial_r^2 \Theta_k + 3\Theta_k \partial_r \Theta_k + \Theta_k^3 = 0 \qquad \Rightarrow \qquad \Theta_k = \frac{2\tau_1(r + \tau_2)}{3 + \tau_1(r + \tau_2)^2}, \quad (3.14c)$$

where the $\tau_i$'s are radial constants.

This shows that the two differential gauge conditions (3.1) and (3.8) can also be understood as geometrical conditions on the null rigging vector, involving its inaffinity for the NU gauge and its expansion for the BS gauge.

## 4 Charges

We now turn to the study of the asymptotic charges in the partial Bondi gauge. We first compute the "bare" charges, which contain a divergent contribution, and then show that they can be renormalized using the symplectic potential studied in section 2.3. We then perform a consistency check by comparing our result in the limiting case of GBMS with the charges derived in [16, 17, 20], where the metric $q_{AB}$ is constrained to have a fixed determinant but is free to fluctuate otherwise. Finally, as a simple application of our formula for the charges (and as a further consistency check), we compute the charges of the Kerr solution using the partial Bondi gauge metric of [32].

### 4.1 Bare charges

In order to compute the charges we first choose the Barnich–Brandt prescription [78, 79], according to which the charge aspect is given by $\not\!\delta Q_{\text{BB}} = k_{\text{BB}}^{ur}$ with

$$k_{\text{BB}}^{\mu\nu} = \sqrt{-g}\left(2\xi^{[\mu}\nabla^{\nu]}\delta g - 2\xi^{[\mu}\nabla_\alpha\delta g^{\nu]\alpha} + 2\xi_\alpha\nabla^{[\mu}\delta g^{\nu]\alpha} + \delta g\nabla^{[\mu}\xi^{\nu]} + \delta g^{[\mu\alpha}\left(\nabla^{\nu]}\xi_\alpha - \nabla_\alpha\xi^{\nu]}\right)\right). \quad (4.1)$$

Here $[\mu\nu] = (\mu\nu - \nu\mu)/2$ and the variations are $\delta g^{\mu\nu} = \delta(g^{\mu\nu})$ and $\delta g = g_{\mu\nu}\delta g^{\mu\nu} = -\delta\ln g$. We find that the expansion of the charges takes the form

$$\not\!\delta Q_{\text{BB}} = r\not\!\delta Q_\xi^{\text{div}} + \not\!\delta Q_\xi^{\text{finite}} + \mathcal{O}(r^{-1}), \quad (4.2)$$

where the divergent part is

$$\not\!\delta Q_\xi^{\text{div}} = f\left(\frac{1}{2}\sqrt{q}\,\partial_u C^{AB}\delta q_{AB} - \delta\left(\sqrt{q}R\right) - \delta\sqrt{q}\,\partial_u C\right) - (2k + \Delta f)\delta\sqrt{q} - 2Y^A\delta\left(\sqrt{q}U_A^2\right)$$
$$+ \frac{1}{2}h\sqrt{q}\left(q^{AB}\delta C_{AB} - \delta C\right), \quad (4.3)$$

and the finite part is

$$\not\!\delta Q_\xi^{\text{finite}} = \not\!\delta Q_Y + \not\!\delta Q_h + \not\!\delta Q_k + \not\!\delta Q_\ell + \not\!\delta Q_f, \quad (4.4\text{a})$$

$$\not\!\delta Q_Y = Y^A\delta\left[\sqrt{q}\left(2\mathcal{P}_A - \frac{3}{16}\partial_A(4D - [CC]) + C_{AB}U_2^B - CU_A^2\right)\right], \quad (4.4\text{b})$$

$$\not\!\delta Q_h = h\delta\left[\sqrt{q}\left(\frac{3}{2}D + \frac{1}{4}C^2 - \frac{5}{8}[CC]\right)\right], \quad (4.4\text{c})$$

$$\not\!\delta Q_k = \frac{1}{2}k\left(\sqrt{q}\,C_{\text{TF}}^{AB}\delta q_{AB} - C\,\delta\sqrt{q}\right), \quad (4.4\text{d})$$

$$\not\!\delta Q_\ell = -3\ell\delta\sqrt{q}, \quad (4.4\text{e})$$

$$\not\!\delta Q_f = 4f\delta\left(\sqrt{q}\mathcal{M}\right) - \frac{1}{2}f\sqrt{q}\,C_{AB}^{\text{TF}}\delta N^{AB} - \frac{1}{4}fC\delta\left(\sqrt{q}R\right)$$
$$+ \sqrt{q}\,\delta q_{AB}\left[f\left(D^AU_2^B + \frac{1}{4}RC^{AB} + \frac{1}{8}\partial_u CC^{AB} + \frac{1}{8}CN^{AB}\right) + 2\partial^A f U_2^B + \frac{1}{4}\Delta f\,C_{\text{TF}}^{AB}\right]$$
$$+ \delta\sqrt{q}\left[f\left(2\mathcal{M} - \frac{3}{4}\partial_u D - \frac{3}{16}\partial_u[CC] + \frac{1}{8}\partial_u C^2 - 2D_A U_2^A\right) - 4U_2^A\partial_A f - \frac{1}{4}C\Delta f\right]$$
$$+ \sqrt{q}\,D_A\left(f\delta U_2^A + 2f U_2^A\delta\ln\sqrt{q}\right). \quad (4.4\text{f})$$

One can see as announced that the three free functions $(h, k, \ell)$ in the radial part of the vector field are associated with non-vanishing charges, in addition to the usual contributions involving $f$ and $Y$. This is one of the main results of the present work. In particular, $k$ appears

as soon as we work with GBMS and consider $\delta q_{AB} \neq 0$ (with $\sqrt{q}$ fixed), while $\ell$ requires in addition that we allow for $\delta \sqrt{q} \neq 0$ as in BMSW [13, 21].

The expressions in (4.4) are also valid for field-dependent parameters $(Y^A, h, k, \ell, f)$ since we have not commuted any of the variations $\delta$ with these functions. This will be useful later on in section 5 when considering field-dependent redefinitions corresponding to changes of slicings. Importantly, one should note that even for field-independent symmetry parameters the charges contain many non-integrable contributions, which are not only sourced by the news $N_{AB}$. This is of course already the case when working with $\delta q_{AB} \neq 0$ and $\delta \sqrt{q} \neq 0$, but here we see that in the partial Bondi gauge (or in any complete gauge fixing allowing for free traces as discussed above) also $C$ and $D$ contribute to the non-integrability. This non-integrability reflects the fact that the symplectic potential (2.28) contains symplectic pairs in addition to that formed by the news and the shear (i.e. the Ashtekar–Streubel pair). Due to this complicated structure we will not discuss here the full-fledged Wald–Zoupas prescription for conserved integrable charges [48, 80–83] (which would require to study the flux, the notion of stationarity, and that of covariance). We will however take preliminary steps in this direction in section 5 below.

Finally, let us point out that the same result as (4.2) is found using the Iyer–Wald prescription [84], according to which the charges are given in terms of the Noether–Komar charge and the symplectic potential by the $(ur)$ component of

$$k_{\text{IW}}^{\mu\nu} = \delta K_\xi^{\mu\nu} - K_{\delta\xi}^{\mu\nu} + \xi^{[\mu}\theta^{\nu]}, \tag{4.5}$$

where $K_\xi^{\mu\nu} = -\sqrt{-g}\,\nabla^{[\mu}\xi^{\nu]}$ is the Noether–Komar contribution arising from the Einstein–Hilbert Lagrangian and the potential (2.24). There is therefore no ambiguity at this stage, and cohomological as well as covariant phase space methods both lead to (4.4).

## 4.2 Renormalization

The charges (4.2) contain the $r$-divergent contribution (4.3). This divergent contribution can be renormalized with the corner term inherited from the divergent part (2.26) of the radial symplectic potential, or equivalently from $\theta^u$ by virtue of (2.27) [20, 85–87]. Performing an innocent integration by parts on $\delta$, this corner term can be chosen as

$$\vartheta_{\text{ren}} = \frac{1}{2}\sqrt{q}\big(q^{AB}\delta C_{AB}^{\text{TF}} + \delta C\big). \tag{4.6}$$

To show that this corner potential does indeed renormalize the charges (4.2), one can compute its contribution to the charges. Using the transformation laws (2.16) and (2.17), we find[4]

$$\xi \lrcorner (\delta\vartheta_{\text{ren}}) = \oint Q_\xi^{\text{div}} + \sqrt{q}\,D_A\Upsilon^A, \tag{4.7}$$

where the boundary term is

$$\Upsilon^A = \frac{1}{2}Y^A\delta C + Y^A C_{\text{TF}}^{BC}\delta q_{BC} - Y^C\left(\frac{1}{2}\delta q^{AB}C_{BC} + \frac{1}{2}C^{AB}\delta q_{BC} + q^{AB}\delta C_{BC}^{\text{TF}}\right)$$
$$- f D_B\delta q^{AB} + \delta q^{AB}\partial_B f - 2f\,\partial^A\delta\ln\sqrt{q} + 2\partial^A f\,\delta\ln\sqrt{q}. \tag{4.8}$$

As expected, the renormalization via a corner term works in the same way as in the case of GBMS or BMSW. It would be interesting to obtain this corner term as the symplectic potential of a boundary Lagrangian, following the prescription of [44, 45, 48, 88, 89], but we leave this investigation for future work.

---

[4] Here $\xi\lrcorner$ denotes the contraction in field space between a variational vector field $\delta_\xi$ and a variational 1-form. For example $\xi\lrcorner\delta Q = \delta_\xi Q$. The notations $\xi\lrcorner$ and $\xi\lrcorner$ are sometimes denoted by $I_\xi$ and $\iota_\xi$ respectively.

### 4.3 Comparison with the literature

As a consistency check for the involved expression (4.2), we can now compare the results with the expressions given in [20] for GBMS. For this we reduce the solution space by setting

$$C = 0, \qquad D = \frac{1}{2}[CC], \qquad \delta\sqrt{q} = 0, \qquad N_A = -\frac{2}{3}N_A^{\text{BT}} + \frac{1}{3}C_{AB}D_C C^{BC}, \quad (4.9)$$

which accounts for the fact that the setup of [20] is the Bondi–Sachs gauge with a fixed volume element and with a different definition of the bare angular momentum. With this, the symmetry parameters in BS gauge become

$$h = -\frac{1}{2}D_A Y^A, \qquad k = 0, \qquad \ell = -\frac{1}{4}C^{AB}D_A\partial_B f. \quad (4.10)$$

Recalling that when $C = 0$ and $\delta\sqrt{q} = 0$ we have $C_{AB}^{\text{TF}} = C_{AB}$, $\delta[CC] = 2C^{AB}\delta C_{AB}$, and $\delta(D_A V^A) = D_A(\delta V^A)$, we find that the finite part of the charge reduces to

$$\cancel{\oint}Q_Y = Y^A\delta\left[\sqrt{q}\left(2\mathcal{P}_A - \frac{3}{16}\partial_A[CC] + C_{AB}U_2^B\right)\right] = 2Y^A\delta(\sqrt{q}N_A^{\text{BT}}), \quad (4.11a)$$

$$\cancel{\oint}Q_h = \frac{1}{8}h\delta(\sqrt{q}[CC]) = \frac{1}{16}Y^A\delta(\sqrt{q}\,\partial_A[CC]) - \frac{1}{16}\sqrt{q}D_A(Y^A\delta[CC]), \quad (4.11b)$$

$$\cancel{\oint}Q_k = 0, \quad (4.11c)$$

$$\cancel{\oint}Q_\ell = 0, \quad (4.11d)$$

$$\cancel{\oint}Q_f = 4f\sqrt{q}\,\delta\mathcal{M} - \frac{1}{2}f\sqrt{q}\,C_{AB}\delta N^{AB} + \sqrt{q}D_A(f\,\delta U_2^A)$$
$$+ \sqrt{q}\,\delta q_{AB}\left[f\left(D^A U_2^B + \frac{1}{4}RC^{AB}\right) + 2\partial^A f U_2^B + \frac{1}{4}\Delta f\,C^{AB}\right]. \quad (4.11e)$$

These are exactly the ingredients of equation (5.31) of [20]. Further imposing $\delta q_{AB} = 0$ then leads to the result of [12], in which the split between integrable charge and flux was chosen as $\delta Q_\xi + \Xi_\xi[\delta]$ with

$$Q_\xi \overset{\circ}{=} \sqrt{q}\left[4fM + Y^A\left(2N_A^{\text{BT}} + \frac{1}{16}\partial_A[CC]\right)\right], \qquad \Xi_\xi[\delta] = \frac{1}{2}f\sqrt{q}N^{AB}\delta C_{AB}, \quad (4.12)$$

where $\overset{\circ}{=}$ denotes an equality which holds modulo total divergencies on the 2-sphere (i.e. terms which drop when integrating the charge aspect).

### 4.4 Kerr in the partial Bondi gauge

As explained above, the symmetry parameters $(h, k, \ell)$ appear in the charges only when the induced boundary metric is such that $\delta q_{AB} \neq 0 \neq \delta\sqrt{q}$. However, our analysis brings a novelty even in the case where the boundary metric is completely fixed, namely an expression for the charges valid in the partial Bondi gauge. This enables to compute for example the charges in Newman–Unti gauge for "standard" asymptotically-flat boundary conditions, i.e for fixed boundary data. Another interesting application is to compute the charges for the Kerr metric in Bondi gauge. Indeed, as explained in [32], when writing the Kerr metric in Bondi gauge one is first naturally led to what the authors of this reference call "generalized Bondi–Sachs" coordinates. These represent an example of a partial Bondi gauge. While it is of course possible to then perform a redefinition of the radial coordinate in order to bring the metric in Bondi–Sachs gauge (i.e. such that the BS determinant condition is satisfied), this step becomes superfluous once an expression for the charges in the partial Bondi gauge is available. This is essentially what (4.4) provides.

With this motivation in mind, let us consider a subsector of the general solution space studied above, obtained by setting $\delta q_{AB} = 0$ while staying within the partial Bondi gauge. In this case, using the fact that when $\delta\sqrt{q} = 0$ we have $2h = -D_A Y^A$ and $\delta(D_A V^A) = D_A(\delta V^A)$, the finite charge (4.4) reduces to

$$\oint Q^{\text{finite}} \stackrel{\circ}{=} Y^A \sqrt{q}\, \delta\left(-3N_A + \frac{1}{16}\partial_A(12D - 5[CC] - 2C^2) + C_{AB}D_C C^{BC} + \frac{1}{2}D^B(CC_{AB}) - \frac{3}{2}C_{AB}\partial^B C\right)$$
$$+ 4f\sqrt{q}\,\delta\mathcal{M} - \frac{1}{2}f\sqrt{q}\,C_{AB}^{\text{TF}}\delta N^{AB}, \tag{4.13}$$

where as indicated by the equality $\stackrel{\circ}{=}$ we have dropped a total derivative. Importantly, it should be noted that the traces $C$ and $D$ still appear in this expression.

The Kerr metric in generalized Bondi–Sachs coordinates $(u, r, \theta, \phi)$ can be found in [32] (see also [33,34]). With our choice of mostly plus signature, the asymptotic form of this metric is given by the non-vanishing components

$$g_{uu} = -1 + \frac{2m}{r} + \mathcal{O}(r^{-2}), \tag{4.14a}$$

$$g_{ur} = -1 + \frac{a^2}{r^2}\left(\frac{1}{2} - \cos^2\theta\right) + \mathcal{O}(r^{-3}), \tag{4.14b}$$

$$g_{u\theta} = -a\cos\theta + \frac{2}{r}a(m - a\sin\theta)\cos\theta + \mathcal{O}(r^{-2}), \tag{4.14c}$$

$$g_{u\phi} = -\frac{2}{r}am\sin^2\theta + \mathcal{O}(r^{-2}), \tag{4.14d}$$

$$g_{\theta\phi} = -\frac{2}{r}a^2 m\sin^2\theta\cos\theta + \mathcal{O}(r^{-2}), \tag{4.14e}$$

$$g_{\theta\theta} = r^2 + 2ra\sin\theta + a^2(3\sin^2\theta - 1) + \mathcal{O}(r^{-1}), \tag{4.14f}$$

$$g_{\phi\phi} = r^2\sin^2\theta - 2ra\sin\theta\cos^2\theta + a^2(1 - 3\sin^2\theta\cos^2\theta) + \mathcal{O}(r^{-1}), \tag{4.14g}$$

where the parameters are $m$ and $a$. From this we can then extract the data appearing in the solution space described in section 2.1. For the tensors in the transverse metric we find

$$q_{AB}\,\mathrm{d}x^A\mathrm{d}x^B = \mathrm{d}\theta^2 + \sin^2\theta\,\mathrm{d}\phi^2, \tag{4.15a}$$

$$C_{AB}\,\mathrm{d}x^A\mathrm{d}x^B = 2a\sin\theta\,\mathrm{d}\theta^2 - 2a\sin\theta\cos^2\theta\,\mathrm{d}\phi^2, \tag{4.15b}$$

$$D_{AB}\,\mathrm{d}x^A\mathrm{d}x^B = a^2(3\sin^2\theta - 1)\mathrm{d}\theta^2 + a^2(1 - 3\sin^2\theta\cos^2\theta)\mathrm{d}\phi^2. \tag{4.15c}$$

One can then read in $g_{uu}$ that the mass is $M = m$. Finally the bare angular momentum, which can most easily be extracted from the inverse metric using

$$g^{rA} = -\frac{U_2^A}{r^2} - \frac{N^A}{r^3} + \mathcal{O}(r^{-4}), \tag{4.16}$$

is given by

$$N^A\partial_A = 2am(\partial_\phi - \cos\theta\,\partial_\theta), \qquad N_A\mathrm{d}x^A = 2am(\sin^2\theta\,\mathrm{d}\phi - \cos\theta\,\mathrm{d}\theta). \tag{4.17}$$

As expected, $q_{AB}$ is the round sphere metric. However, while the line element (4.14) is clearly in the partial Bondi gauge, one can see that the trace of $C_{AB}$ is non-vanishing and given instead by $C = q^{AB}C_{AB} = -2a\cos(2\theta)\csc\theta$. The line element is therefore not in BS gauge. From (4.14b) one can furthermore see that $\beta \neq 0$, meaning that the line element is not in NU gauge either. One possibility to deal with this is to redefine the radial coordinate so as to bring the line element in BS gauge, as was done in [12]. However the result (4.13) enables us to bypass

this step and to directly compute the charges using the line element (4.14). Since $\partial_u C_{AB} = 0$, the charge aspects are integrable and reduce to $\oint Q^{\text{Kerr}} = \delta Q^{\text{Kerr}}$ with

$$Q^{\text{Kerr}} = 2a\big((a + 3m\sin\theta)\cos\theta\, Y^\theta(x^A) - 3m\sin^3\theta\, Y^\phi(x^A)\big) + 4m\sin\theta\, T(x^A). \qquad (4.18)$$

The smeared charges associated to the exact Killing vectors $\partial_u$ and $\partial_\phi$ are then obtained respectively for $(T = 1, Y^\theta = 0, Y^\phi = 0)$ and $(T = 0, Y^\theta = 0, Y^\phi = 1)$, and are given by

$$\mathcal{Q}_{\partial_u}^{\text{Kerr}} = \int_0^\pi \mathrm{d}\theta \int_0^{2\pi} \mathrm{d}\phi\, Q_{\partial_u}^{\text{Kerr}} = 16\pi m, \qquad \mathcal{Q}_{\partial_\phi}^{\text{Kerr}} = \int_0^\pi \mathrm{d}\theta \int_0^{2\pi} \mathrm{d}\phi\, Q_{\partial_\phi}^{\text{Kerr}} = -16\pi am. \tag{4.19}$$

Up to a normalization factor of $16\pi G$ which we have dropped from the onset, these are the expected results.

# 5 Charge algebra with conformal boundary metric

We now turn to the study of the charge algebra. This is a subtle task because of the complicated expression (4.4) for the charges, and especially because these contain many non-integrable contributions in addition to the one arising from the news. This non-integrability is also intimately tied to the choice of charge bracket, since proposals such as the Barnich–Troessaert bracket [12] require a split between integrable and non-integrable parts. While for standard asymptotically-flat boundary conditions with $\delta q_{AB} = 0$ the split can be singled out by the Wald–Zoupas criterion [80, 83], an extension of the prescription to the case $\delta q_{AB} \neq 0$ is still missing. Setting this issue aside, here we aim at showing that under the Barnich–Troessaert bracket the charges represent the symmetry algebra (2.22) up to a field-dependent 2-cocycle. We will choose the integrable part of the charges to be conserved in the radiative vacuum where $\mathcal{J}^A = 0 = \mathcal{N}^{AB}$. We will then show that the modified prescription of [70] (which we will refer to as the Koszul bracket) leads to a vanishing cocycle.

In order to compute the algebra of the charge (4.2), we will consider for simplicity the conformal gauge for the boundary metric. This is defined by the choice $q_{AB} = e^\Phi q_{AB}^\circ$, so that $\delta q_{AB} = q_{AB}\delta\Phi$ and therefore

$$\delta\Phi = \delta\ln\sqrt{q}, \qquad \delta\Gamma_{AB}^C = \frac{1}{2}\big(\delta_B^C\partial_A + \delta_A^C\partial_B - q_{AB}^\circ\partial^C\big)\delta\Phi, \qquad \delta R = -\delta\Phi R - \Delta\delta\Phi. \tag{5.1}$$

From the transformation laws (2.10) and (2.16a), one can see that using the above conformal boundary metric requires to impose the consistency condition

$$\delta_\xi q_{AB} = D_A Y_B + D_B Y_A + 2h q_{AB} \stackrel{!}{=} q_{AB}\delta_\xi\Phi = q_{AB}\big(D_A Y^A + 2h\big)$$
$$\Rightarrow D_A Y_B + D_B Y_A = q_{AB}(D_C Y^C), \tag{5.2}$$

which therefore means that $Y^A$ has to satisfy the conformal Killing equation.

When using the conformal gauge for the boundary metric, the contribution (4.4d) of $k$ to the charge becomes proportional to $k C\delta\sqrt{q}$. This contribution can then be removed with a field-dependent redefinition of the function $\ell$ in (4.4e), also known as a change of slicing [42, 59, 79]. At the end of the day, this therefore means that in the conformal gauge $k$ can be viewed as pure gauge, so that one can set $C = 0$ and $C_{AB}^{\text{TF}} = C_{AB}$. Let us now introduce for convenience the objects[5]

$$\tilde{D} := -\frac{3}{8}D + \frac{5}{32}[CC], \qquad C_A := \frac{1}{4}\partial_A[CC] + C_{AB}D_C C^{CB}, \tag{5.3}$$

---

[5]The transformation of $\tilde{D}$ is given in (5.8), while that of $C_A$ is derived in appendix A.

and perform a change of slicing by substituting the parameter $\ell$ by $\tilde{\ell}$ defined as

$$\tilde{\ell} := -\frac{3}{2}\ell + f\,\partial_u\tilde{D} - \frac{1}{4}D_A\partial_B f\,C^{AB}, \tag{5.4}$$

where now $\delta\tilde{\ell} = 0$. Let us also write the symmetry parameter $f$ as $f(u, x^A) = T(x^A) - uh(x^A)$. Using $C_{AB}\delta N^{AB} = C^{AB}\delta N_{AB} - 2[CN]\delta\Phi$ and discarding total derivative terms, the finite part of the charge (4.4) finally becomes[6]

$$\oint Q_\xi = \delta Q_\xi^{\mathrm{int}} + \Xi_\xi[\delta], \tag{5.6}$$

where

$$Q_\xi^{\mathrm{int}} = 2\sqrt{q}\Big(Y^A\big(\mathcal{P}_A - u\partial_A\mathcal{M} + \partial_A\tilde{D}\big) + 2T\mathcal{M} + \tilde{\ell} - 2h\tilde{D}\Big), \tag{5.7a}$$

$$\begin{aligned}
\Xi_\xi[\delta] \overset{\circ}{=}\; &-\frac{1}{2}f\sqrt{q}\,C^{AB}\delta N_{AB} - \frac{1}{2}Y^A\delta\big(\sqrt{q}\,C_A\big) \\
&+ \delta\sqrt{q}\Big(2f\mathcal{M} + 2uD_A(Y^A\mathcal{M}) + \frac{1}{2}D_AD_B\big(f\,C^{AB}\big)\Big) - 2u\big(2h + D_AY^A\big)\delta\big(\sqrt{q}\,\mathcal{M}\big).
\end{aligned} \tag{5.7b}$$

The notation $\overset{\circ}{=}$ means that for this equality we have dropped a total derivative, i.e. a term which would vanish upon integration over the celestial sphere. The motivation for this split is that the integrable part is conserved in the vacuum defined by the conditions $\mathcal{J}^A = 0 = \mathcal{N}^{AB}$, provided we impose as well that $\partial_u\tilde{D} = 0 = \partial_u\tilde{\ell}$. These two conditions are compatible since $\tilde{D}$ transforms as

$$\delta_\xi\tilde{D} = (\mathcal{L}_Y - 2h)\tilde{D} + \tilde{\ell}. \tag{5.8}$$

One can easily check the conservation of the integrable part of the charge using the evolution equations (2.6). Moreover, after the change of slicing defined by (5.4), for field-independent functions $(T, h, \tilde{\ell}, Y)$ the vector fields still form an algebra. The commutation relations are

$$\big[\xi(T_1, h_1, \tilde{\ell}_1, Y_1), \xi(T_2, h_2, \tilde{\ell}_2, Y_2)\big]_* = \xi(T_{12}, h_{12}, \tilde{\ell}_{12}, Y_{12}), \tag{5.9}$$

with

$$T_{12} = -T_1h_2 + Y_1^A\partial_A T_2 - (1 \leftrightarrow 2), \tag{5.10a}$$

$$h_{12} = Y_1^A\partial_A h_2 - (1 \leftrightarrow 2), \tag{5.10b}$$

$$\tilde{\ell}_{12} = Y_1^A\partial_A\tilde{\ell}_2 - 2h_1\tilde{\ell}_2 - (1 \leftrightarrow 2), \tag{5.10c}$$

$$Y_{12}^A = Y_1^B\partial_B Y_2^A - (1 \leftrightarrow 2). \tag{5.10d}$$

The fact that we obtain an algebra in this slicing is yet another motivation for the above choice of charges (5.7).

We are now ready to compute the algebra of the charges (5.7), first using the Barnich–Troessaert bracket [12], and then the Koszul bracket [69,70] in order to remove the field-dependent central extension. The Barnich–Troessaert bracket is defined as

$$\big\{Q_{\xi_1}^{\mathrm{int}}, Q_{\xi_2}^{\mathrm{int}}\big\}_{\mathrm{BT}} := \delta_{\xi_2}Q_{\xi_1}^{\mathrm{int}} + \Xi_{\xi_2}[\delta_{\xi_1}]. \tag{5.11}$$

---

[6]Let us also mention for completeness that when using the conformal gauge and when setting $C = 0$ the symplectic potential (2.28) reduces to

$$\theta_0^r \overset{\circ}{=} 2\delta\sqrt{q}\Big(\mathcal{M} + \partial_u\tilde{D} + \frac{1}{4}D_AD_B C^{AB}\Big) - \frac{1}{2}\sqrt{q}\,C^{AB}\delta N_{AB} + \delta(\dots). \tag{5.5}$$

A lengthy calculation detailed in appendix B leads to the result

$$\{Q^{\text{int}}_{\xi_1}, Q^{\text{int}}_{\xi_2}\}_{\text{BT}} \stackrel{\circ}{=} Q^{\text{int}}_{\xi_{12}} + \mathcal{K}_{\xi_1,\xi_2}, \tag{5.12}$$

where

$$\begin{aligned}
\mathcal{K}_{\xi_1,\xi_2} &= 2u\sqrt{q}\left(2h_{12} + D_A Y^A_{12} + h_1 D_A Y^A_2 - h_2 D_A Y^A_1\right)\mathcal{M} \\
&\quad + 2u\sqrt{q}\left(\left(2h_1 + D_A Y^A_1\right)\delta_{f_2}\mathcal{M} - \left(2h_2 + D_A Y^A_2\right)\delta_{f_1}\mathcal{M}\right) \\
&\quad + \sqrt{q}\left(2\mathcal{J}^A(f_1 \partial_A f_2 - f_2 \partial_A f_1) + \frac{1}{2}Y^A_{12}C_A + \frac{1}{2}Y^A_1 \delta_{f_2}C_A - \frac{1}{2}Y^A_2 \delta_{f_1}C_A\right). \tag{5.13}
\end{aligned}$$

As expected, one can see that the charges represent the symmetry algebra up to a field-dependent cocycle $\mathcal{K}$ (which is antisymmetric in $1, 2$, as it should). A drawback of the Barnich–Troessaert bracket (5.11) is however that it depends on the choice of split (5.6) between integrable part and flux (we use the term flux loosely since it may contain non-integrable terms other than the news). More precisely, the cocycle changes with redefinitions of the split as in (3.9) of [12]. As mentioned above, one possibility for determining the integrable part is to use the Wald–Zoupas prescription. We defer its study in the case of the partial Bondi gauge to future work.

Let us now consider instead the Koszul bracket of charges introduced in [69,70]. This is an improvement of the Barnich–Troessaert bracket which has precisely the advantage of being independent of the split (5.6) (while still requiring an arbitrary "reference" split to be computed). This new bracket has been used in three-dimensional Einstein–Maxwell theory and shown to lead to a field-dependent central extension in the case of the Wald–Zoupas charges [90]. Once again, here we are going to remain agnostic about the Wald–Zoupas charges, and simply consider the reference split (5.7). The Koszul bracket is then defined as

$$\{Q^{\text{int}}_{\xi_1}, Q^{\text{int}}_{\xi_2}\}_{\text{K}} := \{Q^{\text{int}}_{\xi_1}, Q^{\text{int}}_{\xi_2}\}_{\text{BT}} + \int_\gamma K_{\xi_1,\xi_2}(\Xi), \tag{5.14}$$

where the second term is the integral along a path in field space of the variational 1-form

$$K_{\xi_1,\xi_2}(\Xi) := -\delta_{\xi_1}(\Xi_{\xi_2}[\delta]) + \delta_{\xi_2}(\Xi_{\xi_1}[\delta]) - \Xi_{[\xi_1,\xi_2]_*}[\delta]. \tag{5.15}$$

Since the Barnich–Troessaert bracket has been computed above in (5.12), we only need to compute the additional contribution coming from the variational object (5.15). We defer this tedious calculation to appendix C. The upshot is that with (5.7b) we find

$$K_{\xi_1,\xi_2}(\Xi) = -\delta\mathcal{K}_{\xi_1,\xi_2}, \tag{5.16}$$

where $\mathcal{K}_{\xi_1,\xi_2}$ is the cocycle appearing in (5.12). This implies that the Koszul bracket gives

$$\{Q^{\text{int}}_{\xi_1}, Q^{\text{int}}_{\xi_2}\}_{\text{K}} = Q^{\text{int}}_{\xi_{12}}, \tag{5.17}$$

up to a possible constant central charge due to the field space integration in (5.14).

In summary, we have shown in this section that the algebra of the charges (4.4) in the conformal gauge forms a representation of the asymptotic Killing vector algebra, with or without field-dependent central extension depending on the bracket being used. This is a non-trivial cross-check of our result for the charges.

# 6 Perspectives

In this work we have continued the study of the partial Bondi gauge initiated in the companion paper [1], by focusing on the asymptotic symmetries and charges. We have first recalled in section 2 the structure of the solution space in the partial Bondi gauge, and how the key property of the latter is that it contains free functions of $(u, x^A)$ which are the traces of the tensors appearing in the radial expansion of the transverse metric. We have then shown that the asymptotic Killing vector field contains, correspondingly, free functions of $(u, x^A)$ which are candidates for new symmetry parameters in the asymptotic charges.

In section 3 we have then exhibited new gauge fixing conditions in which only a finite number of these parameters survive as unspecified functions. These new differential Bondi–Sachs and Newman–Unti gauge conditions follow a similar mechanism as the Barnich–Troessaert differential determinant condition [13] which enables to access the Weyl transformations. By increasing the degree of the differential condition, one can access more symmetry transformations while still completely fixing the gauge. One could argue that by doing so one is introducing spurious gauge freedom, but it is the computation of the asymptotic charges which shows precisely what should be considered as gauge or not.

In section 4 we have computed the asymptotic charges, and shown that when allowing for variations $\delta q_{AB} \neq 0 \neq \delta \sqrt{q}$ there are indeed two new asymptotic charges in addition to the charges for super-translations, super-rotations, and Weyl transformations. These two extra charges correspond to the two subleading terms after the Weyl generator in the asymptotic Killing vector field. In fact, one of these extra charges is already genuinely present in the standard Newman–Unti gauge [9]. This has not been noticed previously because it is common in Newman–Unti gauge to fix the origin of the affine parameter, which amounts to setting this charge to zero by hand. As an application of the general expression for the charges in the partial Bondi gauge we have then computed the charges for the Kerr spacetime in Bondi-type coordinates.

In section 5 we have then computed the charge algebra in a simplified setting as a consistency check. For this computation we have used two definitions of the bracket of charges adapted to dealing with their non-integrability, namely the Barnich–Troessaert bracket [12] and the Koszul bracket [69, 70]. The former enables to represent the algebra of asymptotic symmetries up to a field-dependent two-cocycle, while the latter enables to remove this field-dependency.

We now list a few interesting directions in which this work could be developed:

- **Charges for the full partial Bondi gauge solution space.** The first obvious extension of this work would be to compute the asymptotic charges for the general solution space in partial Bondi gauge studied in [1]. This is a daunting task since this general solution space contains all the data allowed by the Einstein equations once the transverse fall-offs have been chosen. This includes a non-vanishing cosmological constant, logarithmic terms, and a free time-dependent boundary metric (parametrized by $\beta_0$, $U_0^A$, and $q_{AB}$). Indeed, recall that in the present work we have turned off all this data appart from a time-independent $q_{AB}$. This is indeed the context which was sufficient in order to reveal the existence of the two new asymptotic charges. It would be reasonable to generalize this study by studying separately the effect of a non-vanishing cosmological constant and of the logarithmic terms. The study of the charges at null infinity in the presence of logarithmic terms and violations of peeling has been initiated in [91]. There is also an existing literature on the charges and the flat limit in (A)dS [18, 19, 92–97], and it would be interesting to study whether the partial Bondi gauge and its extra set of charges could play a role in this context.

- **Wald–Zoupas charges.** In section 5 we have chosen to split the non-integrable charge as (5.7) because the resulting integrable part is conserved in the radiative vacuum. Although this is inspired by the Wald–Zoupas prescription for conserved charges [48, 80–83], we have not worked out the details of the prescription, which requires in principle to introduce a Wald–Zoupas symplectic potential and study the stationarity conditions and the possible anomalies. This is a very important yet subtle issue, because when working in the partial Bondi gauge and allowing for $\delta q_{AB} \neq 0$ many new sources of non-integrability appear in the charges, which reflects the flux terms appearing in the symplectic potential (2.28). Ultimately, this question is presumably related to the physical interpretation of the fields $C \neq 0$ and $D \neq 0$ and of a varying boundary metric $q_{AB}$, which falls outside of the standard framework for gravitational waves emitted by binary systems.

- **Relationship with finite distance corner symmetries.** An important conceptual question is that of the relationship between the asymptotic symmetry algebra (2.23) discovered here and the universal corner symmetry algebra found at finite distance in [51, 52, 54]. The computation presented here, as far as we know, gives so far the largest asymptotic symmetry algebra in four spacetime dimensions. This algebra is $(6 \cdot \infty)$-dimensional, with the 6 components given by super-translations, two super-rotations, Weyl transformations, and the subleading transformations generated by $k$ and $\ell$. This is to be compared with the $(8 \cdot \infty)$-dimensional universal corner symmetry group $G = \text{Diff}(S^2) \ltimes \left(\text{GL}(2, \mathbb{R}) \ltimes \mathbb{R}^2\right)$ found in [51, 52, 54]. It has been argued previously that the $(4 \cdot \infty)$-dimensional BMSW group (obtained by dropping $k$ and $\ell$) corresponds to the asymptotic realization of a subgroup of $G$ [21]. A natural question is therefore whether the extension by $k$ and $\ell$ found in (2.23) is also represented at finite distance in the corner symmetry algebra. If this is the case, one should then investigate whether there is yet another set of gauge and boundary conditions for which the full universal corner symmetry algebra is realized asymptotically, and how exactly this happens. Relatedly, one should also investigate the relationship between the extension of the boundary symmetry algebra presented here, and the analysis of [64], which shows that a generic causal surface in $d$ dimensions has $d + 1$ surface charges. In the three-dimensional case the connection between asymptotic and finite null boundaries was explicitly done in [62]. It would be interesting to extend their result to four dimensions, and also to understand to what extend the prescriptions of [51, 52, 54] and [62, 64] differ.

- **Carrollian and celestial holography.** Finally, it would be interesting to study the role of the new charges found in the partial Bondi gauge in approaches to flat space holography such as Carrollian and celestial holography [98–103]. A related and even more relevant question is already that of the role of the Weyl charges in four-dimensional flat holography, as the connection between celestial holography and asymptotic symmetries is only established so far at the level of the BMS algebra, and not BMSW.

## Acknowledgments

It is our pleasure to thank Adrien Fiorucci and Romain Ruzziconi for numerous discussions, Shahin Sheikh-Jabbari and Vahid Taghiloo for their feedback, and Amitabh Virmani for discussions about the Kerr metric in Bondi gauge.

**Funding information** Research at Perimeter Institute is supported in part by the Government of Canada through the Department of Innovation, Science and Economic Development Canada and by the Province of Ontario through the Ministry of Colleges and Universities.

# A  Computation of $\delta_\xi C_A$

In (5.3) we have defined the object $C_A$. Its transformation law is required in order to compute the charge algebra. Since we compute the latter in the case where $C = 0$ and $k = 0$, let us also focus on this particular case to find the transformation of $C_A$. Our goal is to arrive at formula (A.18) for the transformation under $\delta_f$. The computation uses the definition (5.3) and the transformation law (2.16b) with $C_{AB}^{\text{TF}} = C_{AB}$ since $C = 0$. When $C = 0$ we have $\delta[CC] = 2C^{AB}\delta C_{AB} - 2[CC]\delta \ln \sqrt{q}$. This implies that

$$\begin{aligned}
\delta_\xi[CC] &= 2C^{AB}(f\,\partial_u + \pounds_Y + h)C_{AB} - 4C^{AB}D_A\partial_B f - 2[CC](D_A Y^A + 2h) \\
&= (f\,\partial_u + \pounds_Y - 2h)[CC] - 4C^{AB}D_A\partial_B f,
\end{aligned} \tag{A.1}$$

where for the second equality we have used

$$\begin{aligned}
2C^{AB}\pounds_Y C_{AB} &= 2C^{AB}(Y^C\partial_C C_{AB} + 2C_{AC}\partial_B Y^C) \\
&= 2C^{AB}(Y^C D_C C_{AB} + 2C_{AC}D_B Y^C) \\
&= \pounds_Y[CC] + 4C^{AB}C_{AC}D_B Y^C \\
&= \pounds_Y[CC] + 2[CC]D_A Y^A,
\end{aligned} \tag{A.2}$$

and here we have used $2C_{AC}C_B^C = [CC]q_{AB}$. Using $\partial_A(\pounds_Y[CC]) = \pounds_Y(\partial_A[CC])$, this leads to

$$\begin{aligned}
\delta_\xi\partial_A[CC] &= \partial_A\delta_\xi[CC] \\
&= (f\,\partial_u + \pounds_Y - 2h)\partial_A[CC] - 2[CC]\partial_A h + 2[CN]\partial_A f - 4\partial_A(C^{BC}D_B\partial_C f). \tag{A.3}
\end{aligned}$$

Next, using the identities

$$\delta_\xi\Gamma_{AB}^C = \frac{1}{2}(\delta_B^C D_A + \delta_A^C D_B - q_{AB}D^C)(D_E Y^E + 2h), \tag{A.4a}$$

$$\delta_\xi\Gamma_{AC}^C = \delta_\xi\partial_A\ln\sqrt{q} = \partial_A(D_B Y^B + 2h), \tag{A.4b}$$

and the fact that for conformal Killing vectors satisfying (5.2) we have

$$D_A(\pounds_Y C^{AB}) + 2C^{AB}\partial_A(D_C Y^C) = \pounds_Y(D_A C^{AB}), \tag{A.5}$$

we obtain

$$\begin{aligned}
\delta_\xi(D_A C^{AB}) &= D_A\delta_\xi C^{AB} + \delta_\xi\Gamma_{AC}^C C^{AB} + \delta_\xi\Gamma_{AC}^B C^{AC} \\
&= D_A\delta_\xi C^{AB} + 2C^{AB}\partial_A(D_C Y^C + 2h) \\
&= (f\,\partial_u + \pounds_Y - 3h)D_A C^{AB} + \partial_A f N^{AB} + \partial_A h C^{AB} + \partial^B\Delta f - 2\Delta\partial^B f. \tag{A.6}
\end{aligned}$$

Putting this together, we get

$$\begin{aligned}
\delta_\xi(C_{AB}D_C C^{CB}) &= \delta_\xi C_{AB}D_C C^{CB} + C_{AB}\delta_\xi(D_C C^{CB}) \\
&= (f\,\partial_u + \pounds_Y - 2h)(C_{AB}D_C C^{CB}) + (q_{AB}\Delta f - 2D_A\partial_B f)D_C C^{CB} \\
&\quad + C_{AB}(\partial_C f N^{CB} + \partial^B\Delta f - 2\Delta\partial^B f) + \frac{1}{2}[CC]\partial_A h. \tag{A.7}
\end{aligned}$$

Using (A.3) and (A.7), we can now write the final form of the transformation law for $C_A$, which is

$$\begin{aligned}
\delta_\xi C_A &= (f\,\partial_u + \pounds_Y - 2h)C_A + \frac{1}{2}[CN]\partial_A f + C_{AB}N^{BC}\partial_C f \\
&\quad + D^B(\Delta f C_{AB}) - 2(D_A\partial_B f)D_C C^{CB} - \partial_A(C^{BC}D_B\partial_C f) - 2C_{AB}\Delta\partial^B f. \tag{A.8}
\end{aligned}$$

Note for later use that we also have

$$\Delta\partial^B f = \partial^B \Delta f + \frac{1}{2}R\partial^B f. \tag{A.9}$$

Using the identities

$$C_{AC}N_B^C = \frac{1}{2}q_{AB}[CN] + \frac{1}{2}\big(C_{AC}N_B^C - C_{BC}N_A^C\big)$$
$$= \frac{1}{2}q_{AB}[CN] + 4\varepsilon_{AB}\widetilde{\mathcal{M}} - \big(D_A D_C C_B^C - D_B D_C C_A^C\big), \tag{A.10}$$
$$C_{BC}D_A N^{BC} = C_{AB}D_C N^{BC} + C^{BC}D_B N_{AC}, \tag{A.11}$$

we can now write the transformation of $C_A$ under $f$ as

$$\delta_f C_A = f\partial_u C_A + \frac{1}{2}[CN]\partial_A f + C_{AB}N^{BC}\partial_C f$$
$$+ D^B\big(\Delta f\, C_{AB}\big) - 2\big(D_A\partial_B f\big)D_C C^{CB} - \partial_A\big(C^{BC}D_B\partial_C f\big) - 2C_{AB}\Delta\partial^B f$$
$$= f\left(\frac{1}{2}\partial_A[CN] + C_{AB}D_C N^{CB} + N_{AB}D_C C^{CB}\right) + \frac{1}{2}[CN]\partial_A f + C_{AB}N^{BC}\partial_C f$$
$$+ D^B\big(\Delta f\, C_{AB}\big) - 2\big(D_A\partial_B f\big)D_C C^{CB} - \partial_A\big(C^{BC}D_B\partial_C f\big) - 2C_{AB}\Delta\partial^B f$$
$$= f\left(\frac{1}{2}\partial_A[CN] + C_{AB}D_C N^{CB} + N_{AB}D_C C^{CB}\right) + [CN]\partial_A f$$
$$+ 4\widetilde{\mathcal{M}}\tilde{\partial}_A f - \big(D_A D_B C_C^B - D_C D_B C_A^B\big)\partial_C f$$
$$+ D^B\big(\Delta f\, C_{AB}\big) - 2\big(D_A\partial_B f\big)D_C C^{CB} - \partial_A\big(C^{BC}D_B\partial_C f\big) - 2C_{AB}\Delta\partial^B f. \tag{A.12}$$

The first line of this expression can now be rewritten with the following steps:

$$f\frac{1}{2}\partial_A[CN] + f\,C_{AB}D_C N^{CB} + f\,N_{AB}D_C C^{CB} + [CN]\partial_A f$$
$$= f\frac{1}{2}\partial_A[CN] + D_C\big(f\,C_{AB}N^{CB}\big) - D_C\big(f\,C_{AB}\big)N^{CB} + f\,N_{AB}D_C C^{CB} + [CN]\partial_A f$$
$$= f\frac{1}{2}\partial_A[CN] + D_C\big(f\,C_{AB}N^{CB}\big) - D_C\big(f\,C_{AB}\big)N^{CB} + N_{AB}D_C\big(f\,C^{CB}\big) - \partial_C f\,N_{AB}C^{CB} + [CN]\partial_A f$$
$$= f\frac{1}{2}\partial_A[CN] + D_C\big(f\,C_{AB}N^{CB}\big) - 2D_C\big(f\,C_{AB}\big)N^{CB} + N^{BC}D_A\big(f\,C_{BC}\big) - \partial_C f\,N_{AB}C^{CB} + [CN]\partial_A f$$
$$= f\frac{1}{2}\partial_A[CN] - 2D_C\big(f\,C_{AB}\big)N^{CB} + N^{BC}D_A\big(f\,C_{BC}\big) + [CN]\partial_A f + \partial_C\big(f\,C_{AB}N^{CB}\big) - \partial_C f\,N_{AB}C^{CB}$$
$$= \partial_A\big(f[CN]\big) - 2D_C\big(f\,C_{AB}\big)N^{CB} + N^{BC}D_A\big(f\,C_{BC}\big)$$
$$+ D^B\Big(4f\varepsilon_{AB}\widetilde{\mathcal{M}} - f\big(D_A D_C C_B^C - D_B D_C C_A^C\big)\Big) + D^B f\big(4\varepsilon_{AB}\widetilde{\mathcal{M}} - D_A D_C C_B^C + D_B D_C C_A^C\big)$$
$$= \partial_A\big(f[CN]\big) - 2D_C\big(f\,C_{AB}\big)N^{CB} + N^{BC}D_A\big(f\,C_{BC}\big) + 8\widetilde{\mathcal{M}}\tilde{\partial}_A f + 4f\,\tilde{\partial}_A\widetilde{\mathcal{M}}$$
$$+ D^B\Big(f\big(D_B D_C C_A^C - D_A D_C C_B^C\big)\Big) + D^B f\big(D_B D_C C_A^C - D_A D_C C_B^C\big). \tag{A.13}$$

With this, (A.12) can finally be rewritten as

$$\delta_f C_A = \partial_A\big(f[CN]\big) - 2D_C\big(f\,C_{AB}\big)N^{CB} + N^{BC}D_A\big(f\,C_{BC}\big) + 4\big(3\widetilde{\mathcal{M}}\tilde{\partial}_A f + f\,\tilde{\partial}_A\widetilde{\mathcal{M}}\big)$$
$$+ D^B\Big(f\big(D_B D_C C_A^C - D_A D_C C_B^C\big)\Big) + 2D^B f\big(D_B D_C C_A^C - D_A D_C C_B^C\big)$$
$$+ D^B\big(\Delta f\, C_{AB}\big) - 2\big(D_A\partial_B f\big)D_C C^{CB} - \partial_A\big(C^{BC}D_B\partial_C f\big) - 2C_{AB}\Delta\partial^B f. \tag{A.14}$$

We are now going to use the fact that in the conformal gauge $Y$ has to satisfy the conformal Killing equation (5.2). When integrating by parts (and dropping the total derivative) and using this equation, this implies that

$$\sqrt{q}\, Y_A D_B T_{\text{TF}}^{AB} \overset{\circ}{=} -\sqrt{q}\, D_B Y_A T_{\text{TF}}^{AB} = -\frac{1}{2}\sqrt{q}\,\big(D_C Y^C\big) q_{AB} T_{\text{TF}}^{AB} = 0,\tag{A.15}$$

where $T_{\text{TF}}^{AB}$ is any symmetric and trace-free tensor. We will also use the identity

$$\big(D_B D_A - D_A D_B\big)V^B = R_{AB}V^B = \frac{1}{2}R\,V_A.\tag{A.16}$$

Using these properties we can write

$$\begin{aligned}
\sqrt{q}\, Y_1^A \delta_{f_2} C_A \overset{\circ}{=}\ & \sqrt{q}\, Y_1^A\Big[\partial_A\big(f[CN]\big) - 2D_C\big(f\, C_{AB}\big)N^{CB} + N^{BC}D_A\big(f\, C_{BC}\big)\\
&+ 4\big(3\widetilde{\mathcal{M}}\tilde{\partial}_A f + f\,\tilde{\partial}_A\widetilde{\mathcal{M}}\big) + f_2 C_A^B \partial_B R - D_A D_B D_C\big(f_2 C^{AB}\big)\Big]\\
=\ & \sqrt{q}\, Y_1^A\Big[\partial_A\big(f_2[CN]\big) - 2D_C\big(f_2 C_{AB}N^{CB}\big) + N^{BC}D_A\big(f_2 C_{BC}\big)\\
&+ 4\big(3\widetilde{\mathcal{M}}\tilde{\partial}_A f_2 + f_2\tilde{\partial}_A\widetilde{\mathcal{M}} + f_2 C_{AB}\mathcal{J}^B\big) - D_A D_B D_C\big(f_2 C^{AB}\big)\Big].
\end{aligned}\tag{A.17}$$

This finally leads to

$$\begin{aligned}
\frac{1}{2}\sqrt{q}\, Y_1^A \delta_{f_2} C_A = \sqrt{q}\Bigg[\ & 2Y_1^A\big(3\widetilde{\mathcal{M}}\tilde{\partial}_A f_2 + f_2\tilde{\partial}_A\widetilde{\mathcal{M}} + f_2 C_{AB}\mathcal{J}^B\big) + \frac{1}{2}D_A D_B\big(f_2 C^{AB}\big)\big(D_A Y_1^A\big)\\
&- \frac{1}{2}f_2 C^{AB}Y_1^C D_C N_{AB} - f_2 C^{AB}N_{AC}D_B Y_1^C\Bigg],
\end{aligned}\tag{A.18}$$

which we will use in appendix B below to compute the Barnich–Troessaert bracket.

# B  Computation of the charge algebra

In this appendix we give details about the computation of the Barnich–Troessaert bracket (5.11) for the split (5.7). For the first term in the bracket we find

$$\begin{aligned}
\delta_{\xi_2}Q_{\xi_1}^{\text{int}} = Q_{\xi_{12}}^{\text{int}} &- 2\sqrt{q}\, T_2\big(2h_1 + D_A Y_1^A\big)\mathcal{M} + 2\sqrt{q}\, Y_1^A\big(3\widetilde{\mathcal{M}}\tilde{\partial}_A f_2 + f_2\tilde{\partial}_A\widetilde{\mathcal{M}} + f_2 C_{AB}\mathcal{J}^B\big)\\
&+ 2\sqrt{q}\big(2f_1 + u\big(2h_1 + D_A Y_1^A\big)\big)\delta_{f_2}\mathcal{M},
\end{aligned}\tag{B.1}$$

where the transformation (2.19b) and the equation of motion (2.6b) give

$$\delta_f\mathcal{M} = f\left(\frac{1}{2}D_A\mathcal{J}^A + \frac{1}{4}[CN]\right) + \mathcal{J}^A\partial_A f.\tag{B.2}$$

We then compute the contributions to $\Xi_{\xi_2}[\delta_{\xi_1}]$. For the first line of (5.7b) we find

$$-\frac{1}{2}f_2\sqrt{q}\, C^{AB}\delta_{\xi_1}N_{AB} = -f_2\sqrt{q}\, C^{AB}\left(f_1\mathcal{N}_{AB} + \frac{1}{2}Y_1^C D_C N_{AB} + N_{AC}D_B Y_1^C + D_A\partial_B h_1\right),\tag{B.3a}$$

$$-\frac{1}{2}Y_2^A\delta_{\xi_1}\big(\sqrt{q}\, C_A\big) = \frac{1}{2}\sqrt{q}\big(Y_{12}^A C_A - Y_2^A\delta_{f_1}C_A\big).\tag{B.3b}$$

Then for the second line of (5.7b) we find

$$\begin{aligned}
\delta_{\xi_1}&\sqrt{q}\left(2f_2\mathcal{M} + 2uD_A\big(Y_2^A\mathcal{M}\big) + \frac{1}{2}D_A D_B\big(f_2 C^{AB}\big)\right) - 2u\big(2h_2 + D_A Y_2^A\big)\delta_{\xi_1}\big(\sqrt{q}\,\mathcal{M}\big)\\
&= \sqrt{q}\left(4T_2 h_1\mathcal{M} + 2f_2\mathcal{M}D_A Y_1^A + \frac{1}{2}D_A D_B\big(f_2 C^{AB}\big)\big(2h_1 + D_A Y_1^A\big)\right)\\
&\quad + 2u\sqrt{q}\big(Y_{12}^A\partial_A\mathcal{M} + 2h_{12}\mathcal{M} - Y_2^A\partial_A\big(h_1\mathcal{M}\big)\big) - 2u\big(2h_2 + D_A Y_2^A\big)\sqrt{q}\,\delta_{f_1}\mathcal{M}.
\end{aligned}\tag{B.4}$$

Putting the above contributions together, we find after integrating by parts and dropping the boundary terms that

$$
\begin{aligned}
\{Q^{\text{int}}_{\xi_1}, Q^{\text{int}}_{\xi_2}\}_{\text{BT}} \;\mathring{=}\; & Q^{\text{int}}_{\xi_{12}} + 2u\sqrt{q}\Big(Y^A_{12}\partial_A\mathcal{M} + 3h_{12}\mathcal{M} + \big(Y^A_1 h_2 - Y^A_2 h_1\big)\partial_A\mathcal{M}\Big) \\
& + 2u\sqrt{q}\Big(\big(2h_1 + D_A Y^A_1\big)\delta_{f_2}\mathcal{M} - \big(2h_2 + D_A Y^A_2\big)\delta_{f_1}\mathcal{M}\Big) \\
& + \sqrt{q}\Big(2\mathcal{J}^A(f_1\partial_A f_2 - f_2\partial_A f_1) + \tfrac{1}{2}Y^A_{12}C_A - \tfrac{1}{2}Y^A_2\delta_{f_1}C_A\Big) \\
& + \sqrt{q}\Big[2Y^A_1\big(3\widetilde{\mathcal{M}}\tilde{\partial}_A f_2 + f_2\tilde{\partial}_A\widetilde{\mathcal{M}} + f_2 C_{AB}\mathcal{J}^B\big) + \tfrac{1}{2}D_A D_B\big(f_2 C^{AB}\big)\big(D_A Y^A_1\big) \\
& \qquad - \tfrac{1}{2}f_2 C^{AB}Y^C_1 D_C N_{AB} - f_2 C^{AB}N_{AC}D_B Y^C_1\Big].
\end{aligned} \tag{B.5}
$$

Using (A.18) to rewrite the last square bracket, we obtain

$$
\begin{aligned}
\{Q^{\text{int}}_{\xi_1}, Q^{\text{int}}_{\xi_2}\}_{\text{BT}} \;\mathring{=}\; & Q^{\text{int}}_{\xi_{12}} + 2u\sqrt{q}\Big(Y^A_{12}\partial_A\mathcal{M} + 3h_{12}\mathcal{M} + \big(Y^A_1 h_2 - Y^A_2 h_1\big)\partial_A\mathcal{M}\Big) \\
& + 2u\sqrt{q}\Big(\big(2h_1 + D_A Y^A_1\big)\delta_{f_2}\mathcal{M} - \big(2h_2 + D_A Y^A_2\big)\delta_{f_1}\mathcal{M}\Big) \\
& + \sqrt{q}\Big(2\mathcal{J}^A(f_1\partial_A f_2 - f_2\partial_A f_1) + \tfrac{1}{2}Y^A_{12}C_A + \tfrac{1}{2}Y^A_1\delta_{f_2}C_A - \tfrac{1}{2}Y^A_2\delta_{f_1}C_A\Big).
\end{aligned} \tag{B.6}
$$

Finally, after integrating by parts and dropping the boundary terms we arrive at

$$
\begin{aligned}
\{Q^{\text{int}}_{\xi_1}, Q^{\text{int}}_{\xi_2}\}_{\text{BT}} \;\mathring{=}\; & Q^{\text{int}}_{\xi_{12}} + 2u\sqrt{q}\Big(2h_{12} + D_A Y^A_{12} + h_1 D_A Y^A_2 - h_2 D_A Y^A_1\Big)\mathcal{M} \\
& + 2u\sqrt{q}\Big(\big(2h_1 + D_A Y^A_1\big)\delta_{f_2}\mathcal{M} - \big(2h_2 + D_A Y^A_2\big)\delta_{f_1}\mathcal{M}\Big) \\
& + \sqrt{q}\Big(2\mathcal{J}^A(f_1\partial_A f_2 - f_2\partial_A f_1) + \tfrac{1}{2}Y^A_{12}C_A + \tfrac{1}{2}Y^A_1\delta_{f_2}C_A - \tfrac{1}{2}Y^A_2\delta_{f_1}C_A\Big),
\end{aligned} \tag{B.7}
$$

which is the result (5.12) quoted in the main text.

## C  Computation of $K_{\xi_1,\xi_2}(\Xi)$

In order to compute the contribution of $K_{\xi_1,\xi_2}(\Xi)$ to the Koszul bracket, let us rewrite the flux (5.7b) up to a boundary term as

$$
\Xi_\xi[\delta] \;\mathring{=}\; \tfrac{1}{2}f\,C^{AB}\big(D_A D_B\delta\sqrt{q} - \sqrt{q}\,\delta N_{AB}\big) + 2f\,\mathcal{M}\delta\sqrt{q} - \tfrac{1}{2}\delta\big(\sqrt{q}\,Y^A C_A + 4u\sqrt{q}\,(2h + D_A Y^A)\mathcal{M}\big), \tag{C.1}
$$

and then evaluate (5.15) for each of the terms. Using

$$
\delta\Big[\sqrt{q}\,D_A\big(Y^A_1 Y^B_2 C_B\big)\Big] = \sqrt{q}\,D_A\big(Y^A_1 Y^B_2\delta C_B + Y^A_1 Y^B_2 C_B\delta\ln\sqrt{q}\big), \tag{C.2}
$$

and dropping the total derivatives when using similar formulas, we find the contributions

$$
K_{\xi_1,\xi_2}\big(f\,C^{AB}D_A D_B\delta\sqrt{q}\big) \;\mathring{=}\; 4\sqrt{q}\,f_1 C^{AB}\delta\big(D_A\partial_B h_2\big) - \delta\big(\sqrt{q}\,f_1\partial_A f_2\partial^A R\big) - (1\leftrightarrow 2), \tag{C.3a}
$$

$$
K_{\xi_1,\xi_2}\big(f\sqrt{q}\,C^{AB}\delta N_{AB}\big) \;\mathring{=}\; 4\sqrt{q}\,f_1 C^{AB}\delta\big(D_A\partial_B h_2\big) + 2\delta\big(\sqrt{q}\,f_1\partial_A f_2 D_B N^{AB}\big) - (1\leftrightarrow 2), \tag{C.3b}
$$

$$
K_{\xi_1,\xi_2}\big(f\,\mathcal{M}\delta\sqrt{q}\big) \;\mathring{=}\; 0, \tag{C.3c}
$$

$$
K_{\xi_1,\xi_2}\big(\delta\big[\sqrt{q}\,Y^A C_A\big]\big) \;\mathring{=}\; \delta\Big[\sqrt{q}\,\big(Y^A_{12}C_A + Y^A_1\delta_{f_2}C_A - Y^A_2\delta_{f_1}C_A\big)\Big], \tag{C.3d}
$$

$$
K_{\xi_1,\xi_2}\big(\delta\big[\sqrt{q}\,h\mathcal{M}\big]\big) \;\mathring{=}\; \delta\Big[\sqrt{q}\,\big(h_1\delta_{f_2}\mathcal{M} - h_2\delta_{f_1}\mathcal{M}\big)\Big], \tag{C.3e}
$$

$$
\begin{aligned}
K_{\xi_1,\xi_2}\big(\delta\big[\sqrt{q}\,D_A Y^A\mathcal{M}\big]\big) \;\mathring{=}\; & \delta\Big[\sqrt{q}\,\big(2h_{12}\mathcal{M} + D_A Y^A_{12}\mathcal{M} + h_1 D_A Y^A_2\mathcal{M} - h_2 D_A Y^A_1\mathcal{M} \\
& \qquad + D_A Y^A_1\delta_{f_2}\mathcal{M} - D_A Y^A_2\delta_{f_1}\mathcal{M}\big)\Big].
\end{aligned} \tag{C.3f}
$$

Combining all of these terms to compute $K_{\xi_1,\xi_2}(\Xi)$ with $\Xi$ given by (C.1), we find that the result is a total variation

$$K_{\xi_1,\xi_2}(\Xi) = -\delta\mathcal{K}_{\xi_1,\xi_2}, \tag{C.4}$$

where $\mathcal{K}_{\xi_1,\xi_2}$ is given by (5.13).

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
