# Peer review of "The partial Bondi gauge: Gauge fixings and asymptotic charges"

_SciPost Physics, doi:SciPost Phys. 16, 076 (2024)_

## Round 1 · Referee Report · Anonymous (Referee 1) · 2024-2-5

Strengths

1-calculate physical observables (asymptotic charges, their algebra, their central extensions) for asymptotically flat spacetimes with less restrictive boundary conditions as usual
2-compare their results with existing ones in the literature, thus facilitating the appreciation of new (and known) aspects of their boundary conditions
3-provide simple and pertinent example, namely Kerr spacetimes

Weaknesses

1-rely on the partial Bondi gauge of Ref. [1]; thus the merits of their results depend on the merits of that gauge choice
2-the abstract is a bit lengthy
3-appendix A has some formulas that are a bit hard to digest

Report

The paper continues the quest for the most suitable and physically relevant boundary conditions in 4d asymptotically flat spacetimes. While this is already a worthwhile enterprise from a purely academic perspective, it gains additional significance due to its potential relevance for applications in black hole physics, gravitational wave physics, and potentially flat space holography. The specific set of boundary conditions exploited in this work originate from the authors (Ref. [1] is their prequel) and combine other well-known gauges - Bondi and Newman-Unti gauges - into a single entity. As a consequence of their choices, the authors find more symmetries than in either of these more traditional choices. In the bulk part of their paper, they analyze these symmetries, the associated boundary charges and the algebra generated by them. There are numerous fascinating technical aspects for the experts in the field - relation to Carrollian symmetries, the necessety of charge renormalization, the presence of a field dependent cocycle in the Barnich-Troessaert bracket but its absence in the Kozul bracket, etc. - some of which can also be appreciated by outsiders, especially because they provide a simple application to the Kerr spacetime (though there many of the complications are not present).

Overall, I am happy with the paper as it is (up two two optional recommendations, see below) so that I suggest its publication in SciPost Physics in its present form.

Requested changes

I have no changes that I request, but I have two suggestions:

1-Streamline the text in the abstract to make it slightly shorter. For instance, the first two sentences belong to the introduction, but not necessarily to the abstract. If the abstract started with the sentence " We compute and study the asymptotic charges and their algebra in the partial Bondi gauge introduced in Ref. [1] , by focusing on the flat case with ..." no essential info would be lost and the abstract would already be shorter by more than three lines.

2-Reconsider the presentation of Eqs. (A.12) and (A.13). I guess the numerous equality signs are there to guide readers who wish to check these equations in detail. However, if this is the intention it would be useful to add text what happens between the respective equality signs (otherwise it becomes a bit of a game, "can you spot the five differences"). Alternatively, if the main focus is on the results and not so much on the intermediate steps, the authors could eliminates some of the equalities to make these chain of equations look a bit less daunting.

  • validity: top
  • significance: high
  • originality: good
  • clarity: top
  • formatting: perfect
  • grammar: excellent

Author:  Marc Geiller  on 2024-03-08  [id 4349]

(in reply to Report 1 on 2024-02-05)
Category:
answer to question

We thank the referee for his/her careful reading of the manuscript, and for the suggested improvements of the presentation. We have taken these suggestions into account and implemented the following two changes: - The abstract has been shortened by concatenating the first three sentences. - The lengthy steps in appendix A have been removed, and replaced by the unique equation (A.12) together with the sentence above. This does not alter the content since an interested reader may still obtain this equation by manipulating the various formulas derived above. A new version of the manuscript is attached.

Attachment:

PBcharges.pdf

---

## Round 1 · Referee Report · Anonymous (Referee 2) · 2024-3-3

Strengths

1 - Discussion of a partial gauge fixing for asymptotically flat gravity, its relevance, and relation with historical Bondi and Newman-Unti gauges;
2 - Calculation of asymptotic charges in this context. The new gauge parameters allowed by the gauge relaxation are shown to be related to non-vanishing charges (i.e. new physical observables for asymptotically flat gravity).
3 - Calculation of the charge algebra for different proposal for charge brackets in the case of non-integrability (induced by radiation).
4 - Application of the formalism to a concrete solution of physical importance (the Kerr metric).
5 - Quality of presentation and careful comparison with previous literature.

Weaknesses

None

Report

This paper constitutes the sequel to [1], where the Authors proposed a relaxed gauge fixing (the partial Bondi gauge) with respect to the historical and usual Bondi and Newman-Unti coordinates commonly used in the study of gravitational radiation in asymptotically flat spacetime at null infinity. By definition, the partial Bondi gauge allows for more unconstrained parameters in the asymptotic symmetry diffeomorphisms, which are now shown to be non-trivially charged. This technical effort then leads to a better understanding of what is actually lost by the usual gauge fixations, and contributes to building a more complete and robust picture of the asymptotic structure of gravity in the absence of a cosmological constant and to the discovery of new sets of observables in this context.

Key points of the paper include

(i) a nice discussion of the class of possible completions of the partial Bondi gauge fixing and their implications for the enumeration of free functions in the solution space of gravity;

(ii) the calculation of the charges in the partial Bondi gauge, which shows that the gauge relaxation was not done in vain and gives access to more relevant physical observables, as the new symmetry parameters happen to be related to non-vanishing charges;

(iii) the application of the machinery developed so far to the Kerr black hole, following on from the seminal paper [12], and

(iv) the calculation of the charge algebra for various proposals of an improved Peierls bracket (needed because the charges are essentially non-integrable in the presence of radiation), which interestingly shows that one of the proposals allows the charges to represent the asymptotic symmetry algebra without any field-dependent 2-cocycle.

All these results represent an important completion of the research project started in [1] and constitute a technical and conceptual achievement of undeniable importance, in the current context of interest in gravitational wave physics at astrophysical distances and the construction of the holographic correspondence for asymptotically flat spacetime.

Maintaining the momentum of their previous productions, the Authors here present a very well written paper, in which the motivations, derivations, hypotheses and limitations of their analysis are always clearly explained with many details. This also includes meticulous comparisons with previous literature. The care taken in presenting the results always ensures that, despite the technical nature of the work, the manuscript is always fluent and pleasant to read, and the reasoning easy to follow. In particular, the second section is nicely designed to make the article self-contained and to avoid the reader having to constantly refer to the companion paper. Finally, given the current focus on asymptotically flat gravity, its holographic description and the possible practical applications, the scientific soundness, timeliness and relevance of the results discussed are beyond doubt. I am therefore happy to recommend this paper in its present form for publication in SciPost.

Requested changes

None

---

## Editorial Decision

published